# CERTIFYING DISTRIBUTIONAL ROBUSTNESS USING LIPSCHITZ REGULARISATION

## ABSTRACT

Distributional robust risk (DRR) minimisation has arisen as a flexible and effective framework for machine learning. Approximate solutions based on dualisation have become particularly favorable in addressing the semi-infinite optimisation, and they also provide a certificate of the robustness for the worst-case population loss. However existing methods are restricted to either linear models or very small perturbations, and cannot find the globally optimal solution for restricted nonlinear models such as kernel machines. In this paper we resolve these limitations for a general class of kernel space, and our approach is based on a new upper bound of DRRs using an empirical risk regularised by the Lipschitz constant of the model, e.g., deep neural networks and kernel methods. As an application, we showed that it also provides a certificate for adversarial training, and global solutions can be achieved on product kernel machines in polynomial time.

## 1 INTRODUCTION

Regularised risk minimisation has been the workhorse of learning nonlinear hypotheses such as deep neural networks and kernel machines. Recently, *distributional robust risk* (DRR) minimization has emerged as a promising instance with marked efficacy and flexibility. Instead of perturbing the observed data points, DRRs consider perturbations to the *empirical distribution*, constituting an *ambiguity set* $\mathcal{P}$ that lives in the space of data distributions. Let $\Omega$ be an outcome space with (true) distribution $\mu$, e.g., the joint space of input and output. Given a loss function $\ell$, a model $f$ suffers a loss value $\ell_f(\omega)$ over an outcome $\omega$, and the risk of $f$ under $\mu$ is $\mathrm{risk}_\ell(f, \mu) := \mathrm{E}_\mu[\ell_f]$. In DRR minimisation, a model $f$ is sought that minimises the expectation of loss $\ell$ over an ambiguity set $\mathcal{P}$, i.e., that minimises $\sup_{\nu \in \mathcal{P}} \mathrm{risk}_\ell(f, \nu)$ (Delage & Ye, 2010; Goh & Sim, 2010; Wiesemann et al., 2014). The ambiguity sets can be constructed by moment matching (Bhattacharyya et al., 2005; Farnia & Tse, 2016), divergence balls (Ben-Tal et al., 2013; Duchi et al., 2016; Hu & Hong, 2016), or Wasserstein distance balls (Kantorovitch, 1958). In this work we focus on the last due to its favorable properties in statistics and computation, along with extensive applications in DRR (Esfahani & Kuhn, 2018; Gao & Kleywegt, 2016; Zhao & Guan, 2018).

Despite the generality of DRR, its computational efficiency remains a challenge, since the supremum is over a (typically) uncountably infinite dimension space. Tractable equivalent convex programs can be derived only for a limited range of loss functions along with *linear* hypothesis spaces (Blanchet et al., 2016; El Ghaoui & Lebret, 1997; Shafieezadeh-Abadeh et al., 2015; 2017; Xu et al., 2009a;b). Although Shafieezadeh-Abadeh et al. (2017) developed lifted variants for reproducing kernel Hilbert spaces (RKHS) to accommodate nonlinear hypotheses, the perturbation was applied to $\Phi(\omega)$, where $\Phi$ is the implicit feature map. This still falls short of robustness with respect to distributions over $\Omega$.

A more promising technique for optimizing DRRs over *nonlinear* hypothesis spaces—including deep neural networks and kernel machines—is by dualising it to a form that is amenable to (approximate) optimisation (Esfahani & Kuhn, 2018). The fundamental strong duality result was established independently by Blanchet & Murthy (2019) and Gao & Kleywegt (2016), and has been applied to various tasks such as specification of regularisation parameter (Blanchet et al., 2016), design of transport cost (Blanchet et al., 2017), and selection of ambiguity region size for optimal confidence interval (Duchi et al., 2016). In particular, Sinha et al. (2018) used it to construct an efficiently computable *certificate* on the level of robustness for the worst-case population loss. However, these methods are still subject to marked restrictions when applied to smooth nonlinear models. First, Sinha et al. (2018) restricts the perturbation to be small, which despite the common interest of

imperceptible perturbations, leaves unaddressed the equally interesting regimes of medium to large perturbations; see discussions in Openreview (2018). Moreover, although the global solvability of the inner Lagrangian penalty problem (robust surrogate loss) can be ensured by small enough perturbation, there is no practical procedure to compute the threshold. Finally, the overall optimisation of the nonlinear model is still subject to nonconvexity, precluding tractable global solutions for restricted but still general classes of nonlinear models such as kernel methods.

The first goal of this work, therefore, is to develop a novel certificate on distributional robustness that dispenses with these restrictions (§3). Specifically, we will leverage the McShane-Whitney extension theorem (McShane, 1934; Whitney, 1934) to upper bound DRRs by the empirical risk regularized with the Lipschitz constant of the model $f$, while additionally accounting for the underlying transport cost and the loss $\ell$. The result vastly generalises the vector norm regularisation in *linear* binary classification (Shafieezadeh-Abadeh et al., 2017, Thm. 3.11) to nonlinear models and *extended real-valued* cost functions that encode constraints, along with an arbitrary metric space of labels that is general enough for multiclass problems. Appealing to any magnitude of perturbation, it also enjoys improved computational efficiency compared with the robust surrogate loss in Sinha et al. (2018).

A particularly effective domain to apply this new certificate is adversarial learning (Szegedy et al., 2014), where models are trained to be resilient to malicious distortions on the data. Although Lipschitz regularisation has been a popular recipe for robustness (e.g., Anil et al., 2019; Cisse et al., 2017; Farnia et al., 2019; Gouk et al., 2018; Huster et al., 2018; Scaman & Virmaux, 2018) and generalisation accuracy (Miyato et al., 2018; Yoshida & Miyato, 2017), it remains a heuristic and therefore our second major contribution is to reveal in §3.1 that adversarial risks (Goodfellow et al., 2015; Madry et al., 2018; Shaham et al., 2018) can be bounded by a DRR. Such a rigorous justification has hitherto been restricted to logistic loss (Suggala et al., 2019, Thm. 9), and a similar tightness result has been established only for linear models (Shafieezadeh-Abadeh et al., 2017, Thm. 3.20). As a result, our new certificate amounts to a new bound on the worst-case risk under attacks, complementing the existing certificates (Raghunathan et al., 2018; Tsuzuku et al., 2018; Weng et al., 2018; Wong & Kolter, 2018; Wong et al., 2018) with a more computationally efficient approach. It further achieved state-of-the-art accuracy under a range of attacks on standard benchmark datasets (§5).

In practice, however, the evaluation of Lipschitz constant $L$ is NP-hard for neural networks (Scaman & Virmaux, 2018), compelling approximations of it, or explicit engineering of layers to respect Lipschitz while analyzing the expressiveness in specific cases (e.g., $\ell_\infty$ norm in Anil et al. (2019)). We, instead, pursue a new path and explore the following question: does there exist a hypothesis space which: **a)** is expressive enough in modeling; **b)** allows the *exact* value of $L$ to be computed *efficiently*; **c)** enforcing the Lipschitz constant leads to a *convex* constraint that renders efficient optimisation.

Interestingly, kernel machines satisfy all these requirements for some kernels. For example, Gaussian kernels are universal, whose RKHS can approximate any continuous function on a compact set in a *uniform* sense (Micchelli et al., 2006). The RKHS of multi-layer inverse kernels compactly encompasses $\ell_1$-regularized neural networks (Shalev-Shwartz et al., 2011), degrading the generalisation performance by only a polynomial constant (Zhang et al., 2016; 2017). Similar results have been conjectured for Gaussian kernels (Shalev-Shwartz et al., 2011). Our third contribution proves that **b)** can be achieved for *product kernels* such as Gaussian kernels with high probability by using the Nyström approximation (Drineas & Mahoney, 2005; Williams & Seeger, 2000), and $\varepsilon$ approximation error of $L$ requires only $O(1/\varepsilon^2)$ samples (§4). Empirically this approximation is also effective for non-product kernels like inverse kernels. Such a sampling based approach also leads to a *single* convex constraint, making it scalable to 60k examples with even an interior-point solver (§5). The convenience in evaluating $L$ renders our certificate of DRR even more favorable than those based on robust surrogate losses (Blanchet & Murthy, 2019; Gao & Kleywegt, 2016; Sinha et al., 2018).

## 2 PRELIMINARIES

The vast majority of our technical results and proofs are deferred to Appendix A, for which theorem-like statements are numbered to be consistent. The *extended real line* is $\bar{\mathbb{R}} := [-\infty, +\infty]$, $\bar{\mathbb{R}}_{\geq 0} := [0, \infty]$, and $[n] := \{1, 2, \dots, n\}$. For topological spaces $X, Y$, the Borel subsets are $\mathscr{B}(X)$, and the Borel probability measures are $\mathfrak{P}(X)$. The *universal sigma algebra* is $\mathscr{U}(X) := \bigcap_{\mu \in \mathfrak{P}(X)} \mathscr{B}_\mu(X)$ where $\mathscr{B}_\mu(X)$ is the completion of the Borel sets with respect to $\mu \in \mathfrak{P}(X)$. Let $\mathscr{L}_0(X, Y)$ denote the Borel measurable mappings $X \to Y$, and $\mathscr{L}_1(X, \mu)$ denote the Borel functions $f \in \mathscr{L}_0(X, \mathbb{R})$ with $\int |f| \, d\mu < \infty$ for $\mu \in \mathfrak{P}(X)$.

For two measures $\mu, \nu \in \mathfrak{P}(\Omega)$ the set of $(\mu, \nu)$-*couplings* is

$$\Pi(\mu, \nu) := \{\pi \in \mathfrak{P}(\Omega \times \Omega) \mid \mu = \int \pi(\,\cdot\,, \mathrm{d}\omega), \ \nu = \int \pi(\mathrm{d}\omega, \,\cdot\,)\}.$$

Let $c \in \mathscr{L}_0(\Omega \times \Omega, \bar{\mathbb{R}})$. The *c-transportation cost* of $\mu, \nu \in \mathfrak{P}(\Omega)$, and *c-transportation cost ball* of radius $r \geq 0$, centred at $\mu \in \mathfrak{P}(\Omega)$ are respectively

$$\mathrm{cost}_c(\mu, \nu) := \inf\left\{\int c\,\mathrm{d}\pi : \pi \in \Pi(\mu, \nu)\right\} \text{ and } \mathrm{B}_c(\mu, r) := \{\nu \in \mathfrak{P}(\Omega) \mid \mathrm{cost}_c(\mu, \nu) \leq r\}. \quad (1)$$

A function $f : \Omega \to \bar{\mathbb{R}}$ is *c-Lipschitz* if there exists $L \geq 0$ such that

$$\forall \omega_1, \omega_2 \in \mathrm{dom}\, f : \ |f(\omega_1) - f(\omega_2)| \leq Lc(\omega_1, \omega_2). \quad (2)$$

The *least c-Lipschitz constant* of $f$ (cf. Cranko et al., 2019) is the infimum over $L \geq 0$ satisfying (2), and is denoted by $\mathrm{lip}_c(f)$, so that when $(X, d)$ is a metric space $\mathrm{lip}_d(f)$ agrees with the usual Lipschitz notion. When $c : X \to \bar{\mathbb{R}}$ (e.g., when $c$ is a norm), we take $c(x, y) := c(x - y)$ for all $x, y \in X$ in (1) and (2).

## 3 CERTIFICATE FOR DISTRIBUTIONAL ROBUSTNESS

While an elegant concept, the DRR suffers from a lack of tractability. That is, in order to effectively minimise it, we first need to be able to compute or estimate it. When the loss function is convex with respect to the input space this is straight-forward, however in general approximations are necessary. Our first contribution is an upper bound for it.

For a function $f : X \to \bar{\mathbb{R}}$ there is another function $\overline{\mathrm{co}}\, f : X \to \bar{\mathbb{R}}$, called the *convex envelope* of $f$. It is the greatest closed convex function that minorises $f$. The quantity $\rho(f) := \sup_{x \in X}(f(x) - \overline{\mathrm{co}}\, f(x))$ was first suggested by Aubin & Ekeland (1976) to quantify the lack of convexity of a function, and has since shown to be of considerable interest for a variety of nonconvex applications (Askari et al., 2019; Kerdreux et al., 2019; Lemaréchal & Renaud, 2001; Udell & Boyd, 2016). When $X = \mathbb{R}^n$ there are well-known ways to compute both $\overline{\mathrm{co}}\, f$ and $\rho(f)$, and a brief discussion on these appears in the appendix (Remark 2 on p. 18). $\rho(f) = 0$ when $f$ is closed convex.

**Theorem 1.** *Assume $X$ is a separable Fréchet space and fix $\mu \in \mathfrak{P}(X)$. Assume $c : X \to \bar{\mathbb{R}}_{\geq 0}$ is sublinear and continuous, and $f \in \mathscr{L}_1(X, \mu)$ is upper semicontinuous. Then for all $r \geq 0$,*

$$\text{DRR} := \sup_{\nu \in \mathrm{B}_c(\mu, r)} \mathrm{risk}_\ell(f, \nu) \ \leq \ r\,\mathrm{lip}_c(\ell_f) + \mathrm{risk}_\ell(f, \mu). \quad (3)$$

*The **tightness** of the bound can be quantified as follows. Let $\Delta(\mu) := r\,\mathrm{lip}_c(\ell_f) + \mathrm{risk}_\ell(f, \mu) - \sup_{\nu \in \mathrm{B}_c(\mu, r)} \mathrm{risk}_\ell(f, \nu)$. If $\mathrm{lip}_c(f) < \infty$ then*

$$\Delta(\mu) \leq r\left(\mathrm{lip}_c(\ell_f) - \left[\mathrm{lip}_c(\overline{\mathrm{co}}\,\ell_f) - \frac{1}{r}\int(\ell_f - \overline{\mathrm{co}}\,\ell_f)\,\mathrm{d}\mu\right]_+\right), \quad (4)$$

*where $[\,\cdot\,]_+ := \max\{\,\cdot\,, 0\}$ and $1/0 := \infty$, so that when $\ell_f$ is closed convex there is **equality** in (3).*

Clearly (4) is tight for convex $\ell_f$. Furthermore, Proposition 1 shows that (4) is also tight for a large family of *nonconvex* functions and distributions — particularly the upper-semicontinuous loss functions on a compact set $X_0 \subseteq X$, with the collection of probability distributions supported on $X_0$.

**Proposition 1.** *Assume $X$ is a separable Fréchet space with $X_0 \subseteq X$. Assume $c : X \to \bar{\mathbb{R}}_{\geq 0}$ is sublinear and continuous, and $\ell_f \in \bigcap_{\mu \in \mathfrak{P}(X_0)} \mathscr{L}_1(X, \mu)$ is upper semicontinuous, has $\mathrm{lip}_c(\ell_f) < \infty$, and attains its maximum on $X_0$. Then for all $r \geq 0$ with $1/0 := \infty$,*

$$\sup_{\mu \in \mathfrak{P}(X_0)} \Delta(\mu) = r\left(\mathrm{lip}_c(\ell_f) - \left[\mathrm{lip}_c(\overline{\mathrm{co}}\,\ell_f) - \frac{1}{r}\rho(\ell_f)\right]_+\right).$$

Theorem 1 subsumes many existing results (viz. Gao & Kleywegt, 2016, Cor. 2 (iv), Cisse et al., 2017, §3.2, Sinha et al., 2018, Shafieezadeh-Abadeh et al., 2017, Thm. 3.20) with a great deal more generality, applying to a very broad family of models, loss functions, and outcome spaces. It is the first time to our knowledge that the slackness in (3) has been characterised tightly.

The extension of Theorem 1 for robust classification in the absence of label noise is straight-forward.

**Corollary 1.** *Assume $X$ is a separable Fréchet space and $Y$ is a topological space. Fix $\mu \in \mathfrak{P}(X \times Y)$. Assume $c : (X \times Y) \times (X \times Y) \to \bar{\mathbb{R}}$ satisfies $c(x, y, x', y') = c_X(x - x')$ whenever*

$y = y'$ and $c(x, y, x', y') = \infty$ whenever $y \neq y'$, where $c_X : X \to \bar{\mathbb{R}}$ is symmetric, sublinear, and continuous, and $f \in \mathscr{L}_1(X \times Y, \mu)$ is upper semicontinuous. Then for all $r \geq 0$ there is (3).

To see the tightness of the bound, if $\mathrm{lip}_c(\ell_f) < \infty$ there is (4), where the closed convex hull is interpreted as $\overline{\mathrm{co}}(\ell_f)(x, y) := \overline{\mathrm{co}}(\ell_f(\,\cdot\,, y))(x)$. If additionally $\ell_f(\,\cdot\,, y)$ is closed convex for all $y \in Y$, there is equality in (3).

### 3.1 DISTRIBUTIONAL ROBUSTNESS AS ADVERSARIAL ROBUSTNESS

We next show how Theorem 1 can be useful for adversarial learning. The following objective function has been proposed to build a robust classifier. Let $X$ and $Y$ be topological spaces, fix $\mu \in \mathfrak{P}(X \times Y)$ and let $d$ be a metric on $X$. The following objective has been proposed (viz Goodfellow et al., 2015; Madry et al., 2018; Shaham et al., 2018) as a means of learning models that are robust to adversarial perturbations

$$\textit{adversarial risk} \ := \ \int \sup_{\tilde{x} \in \mathrm{B}_d(x,r)} \ell_f(\tilde{x}, y)\mu(\mathrm{d}x \times \mathrm{d}y) = \int \sup_{\tilde{\omega} \in \mathrm{B}_{\tilde{d}}(\omega,r)} \ell_f(\tilde{\omega})\mu(\mathrm{d}\omega), \qquad (5)$$

where in the equality we extend $d$ to a metric on $\Omega := X \times Y$ with

$$\tilde{d}((x, y), (x', y')) := d(x, x') + \infty \, [\![ y \neq y' ]\!] \,.$$

We refer to (5) as the **adversarial risk**.

**Theorem 2.** *Assume* $(X, c)$ *is a separable Banach space. Fix* $\mu \in \mathfrak{P}(X)$ *and let* $R_\mu(r) := \{g \in \mathscr{L}_0(X, \mathbb{R}_{\geq 0}) \mid \int g \, \mathrm{d}\mu \leq r\}$. *Then for* $f \in \mathscr{L}_0(\Omega, \bar{\mathbb{R}})$, $r > 0$ *there is*

$$\textit{variable-radius risk} := \sup_{g \in R_\mu(r)} \int \mu(\mathrm{d}\omega) \sup_{\omega' \in \mathrm{B}_c(\omega, g(\omega))} \ell_f(\omega') \leq \sup_{\nu \in \mathrm{B}_c(\mu, r)} \mathrm{risk}_\ell(f, \nu) = DRR. \ (6)$$

*The equality holds in* (6) *if* $\mu$ *is non-atomically concentrated on a compact subset of* $X$, *on which* $f$ *is continuous with the subspace topology.*

We refer to the left-hand side (LHS) of (6) as the **variable-radius risk**. The variable-radius risk has appeared in various forms in similar results, usually formulated using empirical distributions, that is, an average of Dirac masses, (viz. Gao & Kleywegt, 2016; Shafieezadeh-Abadeh et al., 2017). Of course any finite set is compact, and so any empirical distribution satisfies the concentration assumption. Likewise the subspace topology on a finite set is the discrete topology, which makes the continuity assumption trivial.

Both the adversarial risk and the variable-radius risk imply an uncertainty set over a collection of adversaries that may perturb the data. Figure 5 in the appendix (on p. 20) shows the practical difference between the kinds of adversaries in these uncertainty sets. Immediately there is a corollary similar to Corollary 1 for Theorem 2.

It is easy to see that the variable-radius risk upper bounds the adversarial risk (5) by observing that the constant function $g_r \equiv r$ is included in the supremum over $R_\mu(r)$ in (6). As a result,

$$\textit{adversarial risk} \ \leq \ \textit{variable-radius risk} \ \overset{(a)}{\leq} \ DRR \ \overset{(b)}{\leq} \ \textit{Lipschitz regularised risk} \text{ (RHS of (3)), (7)}$$

where $(a)$ is by Theorem 2 and $(b)$ is by Theorem 1.

In general, it is difficult to characterise the tightness of the upper bounds in Theorem 1 and 2. So we resorted to an empirical demonstration that the **sum of all the three gaps** in (7) is relatively low. We randomly generated 100 Gaussian kernel classifiers $f = \sum_{i=1}^{100} \gamma_i k(x^i, \cdot)$, with $x^i$ sampled from the MNIST dataset and $\gamma_i$ sampled uniformly from $[-2, 2]$. The bandwidth was set to the median of pairwise distances. In Figure 1, the $x$-axis is the adversarial risk in (5) where the perturbation $\delta$ is bounded in $\ell_p$ ball and computed by PGD. The $y$-axis is the Lipschitz regularised empirical risk. The scattered dots lie closely to the diagonal, demonstrating that the above bounds are tight in practice.

## 4 PROVABLE LIPSCHITZ REGULARISATION FOR KERNEL METHODS

Theorems 1 and 2 open up a new path to optimising the adversarial risk (5) by Lipschitz regularisation (RHS of (4)), where the upper bounding relationship is established through DRR. In general, however, it is still hard to compute the Lipschitz constant for a nonlinear model. However, we will show that

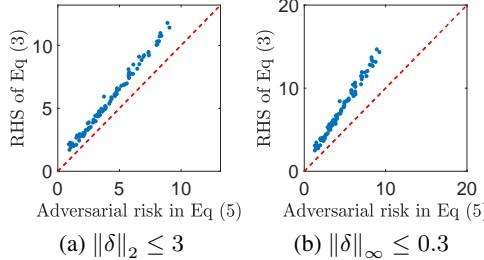

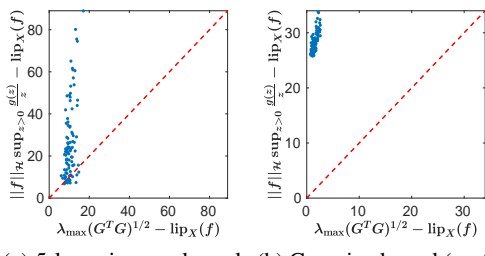

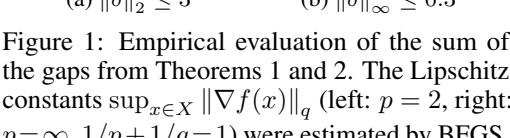

(a) $\|\delta\|_2 \leq 3$     (b) $\|\delta\|_\infty \leq 0.3$

(a) 5-layer inverse kernel    (b) Gaussian kernel ($\sigma=3$)

Figure 1: Empirical evaluation of the sum of the gaps from Theorems 1 and 2. The Lipschitz constants $\sup_{x\in X}\|\nabla f(x)\|_q$ (left: $p=2$, right: $p=\infty$, $1/p+1/q=1$) were estimated by BFGS.

Figure 2: Comparison of $\lambda_{\max}(G^\top G)$ and the RHS of (8), as upper bounds for the Lipschitz constant. Smaller values are tighter. 100 functions sampled in the same way as in Figure 1.

for some types of kernels, this can be done efficiently on functions in its RKHS. Thanks to the known connections between kernel method and deep learning, this technique will also potentially benefit the latter. For example, $\ell_1$-regularised neural networks are compactly contained in the RKHS of multi-layer inverse kernels $k(x,y) = (2 - x^\top y)^{-1}$ with $\|x\|_2 \leq 1$ and $\|y\|_2 \leq 1$ (Zhang et al., 2016, Lemma 1 and Theorem 1) and (Shalev-Shwartz et al., 2011; Zhang et al., 2017), and even possibly Gaussian kernels $k(x,y) = \exp(-\|x-y\|^2/(2\sigma^2))$ (Shalev-Shwartz et al., 2011, §5).

Let us consider a Mercer's kernel $k$ on a convex domain $X \subseteq \mathbb{R}^d$, with the corresponding RKHS denoted as $\mathcal{H}$. The standard kernel method seeks a discriminant function $f$ from $\mathcal{H}$ with the conventional form of finite kernel expansion $f(x) = \frac{1}{l}\sum_{a=1}^l \gamma_a k(x^a, \cdot)$, such that the regularised empirical risk can be minimised with the standard (hinge) loss and RKHS norm. We start with real-valued $f$ for univariate output such as binary classification, and later extend it to multiclass.

**Our goal** here is to additionally enforce, while retaining a convex optimisation in $\gamma := \{\gamma_a\}$, that the Lipschitz constant of $f$ falls below a prescribed threshold $L > 0$, which is equivalent to $\sup_{x\in X}\|\nabla f(x)\|_2 \leq L$ thanks to the convexity of $X$. A quick but primitive solution is to piggyback on the standard RKHS norm constraint $\|f\|_{\mathcal{H}} \leq C$, in view that it already induces an upper bound on $\|\nabla f(x)\|_2$ as shown in Example 3.23 of Shafieezadeh-Abadeh et al. (2017),

$$\sup_{x\in X}\|\nabla f(x)\|_2 \leq \|f\|_{\mathcal{H}}\sup_{z>0} z^{-1}g(z), \quad \text{where} \quad g(z) \geq \sup_{x,x'\in X:\|x-x'\|_2=z}\|k(x,\cdot) - k(x',\cdot)\|_{\mathcal{H}}. \quad (8)$$

For Gaussian kernels, $g(z) = \max\{\sigma^{-1}, 1\}z$. For exponential and inverse kernels, $g(z) = z$ (Bietti & Mairal, 2019). Bietti et al. (2019) justified that the RKHS norm of a neural network may serve as a surrogate for Lipschitz regularisation. But the quality of such an approximation, i.e., the gap in (8), can be loose as we will see later in Figure 2. Besides, $C$ and $L$ are independent parameters.

How can we tighten the approximation? A natural idea is to directly bound the gradient norm at $n$ random locations $\{w^s\}_{s=1}^n$ sampled i.i.d. from $X$. These are obviously convex constraints on $\gamma$. But how many samples are needed in order to ensure $\|\nabla f(x)\|_2 \leq L + \varepsilon$ for *all* $x \in X$? Unfortunately, as shown in Appendix A.1, $n$ may have to grow exponentially by $1/\varepsilon^d$ for a $d$-dimensional space. Therefore we seek a more efficient approach by first slightly relaxing $\|\nabla f(x)\|_2$. Let $g_j(x) := \partial^j f(x)$ be the partial derivative with respect to the $j$-th coordinate of $x$, and $\partial^{i,j} k(x,y)$ be the partial derivative to $x_i$ and $y_j$. $i$ or $j$ being 0 means no derivative. Assuming $\sup_{x\in X} k(x,x) = 1$ and $g_j \in \mathcal{H}$ (true for various kernels considered by Assumptions 1 and 2 below), we get a new bound

$$\sup_{x\in X}\|\nabla f(x)\|_2^2 = \sup_{x\in X}\sum_{j=1}^d \langle g_j, k(x,\cdot)\rangle_{\mathcal{H}}^2 \leq \sup_{\varphi:\|\varphi\|_{\mathcal{H}}=1}\sum_{j=1}^d \langle g_j, \varphi\rangle_{\mathcal{H}}^2 = \lambda_{\max}(G^\top G), \quad (9)$$

where $\lambda_{\max}$ evaluates the maximum eigenvalue, and $G := (g_1, \ldots, g_d)$. The "matrix" is only a notation because each column is a function in $\mathcal{H}$, and obviously the $(i,j)$-th entry of $G^\top G$ is $\langle g_i, g_j\rangle_{\mathcal{H}}$. Interestingly, $\lambda_{\max}(G^\top G)$ delivers *significantly lower* (i.e., tighter) value in approximating the Lipschitz constant $\sup_{x\in X}\|\nabla f(x)\|_2$, compared with $\|f\|_{\mathcal{H}}\max_{z>0}\frac{g(z)}{z}$ from (8). Figure 2 compared these two approximants, where $\lambda_{\max}(G^\top G)$ was computed from (11) derived below, and the landmarks $\{w^s\}$ consisted of all training examples; drawing more samples led to little difference.

Such a positive result motivated us to develop refined algorithms to address the only remaining obstacle to leveraging $\lambda_{\max}(G^\top G)$: no analytic form for computation. Interestingly, it is readily approximable in both theory and practice. Indeed, the role of $g_j$ can be approximated by $\tilde{g}_j$, where $\tilde{g}_j \in \mathbb{R}^n$ is the Nyström approximation (Drineas & Mahoney, 2005; Williams & Seeger, 2000):

$$\tilde{g}_j := K^{-1/2}(g_j(w^1), \ldots, g_j(w^n))^\top = (Z^\top Z)^{-1/2} Z^\top g_j \text{ (noting } g_j(w^1) = \langle g_j, k(w^1, \cdot) \rangle_{\mathcal{H}}) \quad (10)$$

$$\text{where } K := [k(w^i, w^{i'})]_{i,i'}, \quad Z := (k(w^1, \cdot), k(w^2, \cdot), \ldots, k(w^n, \cdot)), \quad \tilde{G} := (\tilde{g}_1, \ldots, \tilde{g}_d).$$

So to ensure $\lambda_{\max}(G^\top G) \leq L^2 + \varepsilon$, intuitively we can resort to enforcing $\lambda_{\max}(\tilde{G}^\top \tilde{G}) \leq L^2$, which also retains the convexity in the constraint in $\boldsymbol{\gamma}$. However, to guarantee $\varepsilon$ error, the number of samples ($n$) required is generally *exponential* (Barron, 1994). Fortunately, we will next show that $n$ can be reduced to *polynomial* for quite a general class of kernels that possess some decomposed structure.

### 4.1 A COORDINATE-WISE NYSTRÖM APPROXIMATION FOR PRODUCT KERNELS

A number of kernels factor multiplicatively over the coordinates, such as periodic kernels (MacKay, 1998), Gaussian kernels, and Laplacian kernels. We will consider $k(x, y) = \prod_{j=1}^d k_0(x_j, y_j)$ where $X = X_0^d$ and $k_0$ is a base kernel on $X_0$. Let the RKHS of $k_0$ be $\mathcal{H}_0$, and let $\mu_0$ be a finite Borel measure with $\operatorname{supp}[\mu_0] = X_0$. Periodic kernels have $k_0(x_j, y_j) = \exp\left(-\sin\left(\frac{\pi}{v}(x_j - y_j)\right)^2/(2\sigma^2)\right)$.

The key benefit of this decomposition is that the derivative $\partial^{0,1} k(x, y)$ can be written as $\partial^{0,1} k_0(x_1, y_1) \prod_{j=2}^d k_0(x_j, y_j)$. Since $k_0(x_j, y_j)$ can be easily dealt with, approximation will be needed *only* for $\partial^{0,1} k_0(x_1, y_1)$. Applying this idea to $g_1 = \frac{1}{l} \sum_{a=1}^l \gamma_a \partial^{0,1} k(x^a, \cdot)$, we can derive

$$\|g_1\|_{\mathcal{H}}^2 = l^{-2} \sum_{a,b=1}^l \gamma_a \gamma_b \langle \partial^{0,1} k_0(x_1^a, \cdot), \partial^{0,1} k_0(x_1^b, \cdot) \rangle_{\mathcal{H}_0} \prod_{j=2}^d k_0(x_j^a, x_j^b), \quad (11)$$

$$\langle g_1, g_2 \rangle_{\mathcal{H}} = l^{-2} \sum_{a,b=1}^l \gamma_a \gamma_b \partial^{0,1} k_0(x_1^a, x_1^b) \partial^{0,1} k_0(x_2^b, x_2^a) \prod_{j=3}^d k_0(x_j^a, x_j^b).$$

So the off-diagonal entries of $G^\top G$ can be computed *exactly*. To approximate the diagonal, we sample $\{w_1^1, \ldots, w_1^n\}$ from $\mu_0$, set $Z_1 = (k_0(w_1^1, \cdot), \ldots, k_0(w_1^n, \cdot))$, and apply Nyström approximation:

$$\langle \partial^{0,1} k_0(x_1^a, \cdot), \partial^{0,1} k_0(x_1^b, \cdot) \rangle_{\mathcal{H}_0} \approx \partial^{0,1} k_0(x_1^a, \cdot)^\top Z_1 \cdot (Z_1^\top Z_1)^{-1} \cdot Z_1^\top \partial^{0,1} k_0(x_1^b, \cdot) \quad (12)$$

$$\text{where} \quad Z_1^\top \partial^{0,1} k_0(x_1^a, \cdot) = (\partial^{0,1} k_0(x_1^a, w_1^1), \ldots, \partial^{0,1} k_0(x_1^a, w_1^n))^\top, \quad (13)$$

and analogously for $Z_1^\top \partial^{0,1} k_0(x_1^b, \cdot)$. We will denote this approximation of $G^\top G$ as $\tilde{P}_G$. Clearly, $\lambda_{\max}(\tilde{P}_G) \leq L^2$ is a convex constraint on $\boldsymbol{\gamma}$, based on i.i.d. samples $\{w_j^s : s \in [n], j \in [d]\}$ from $\mu_0$. It is now important to analyse how many samples $w_j^s$ are needed, such that

$$\lambda_{\max}(\tilde{P}_G) \leq L^2 \quad \Longrightarrow \quad \lambda_{\max}(G^\top G) \leq L^2 + \varepsilon \quad \text{with high probability.}$$

### 4.2 GENERAL SAMPLE COMPLEXITY AND ASSUMPTIONS ON THE PRODUCT KERNEL

Fortunately, product kernels only require approximation bounds for each coordinate, making the sample complexity immune to the exponential growth in the dimensionality $d$. Specifically, we first consider base kernels $k_0$ with a scalar input, i.e., $X_0 \subseteq \mathbb{R}$. Recall from Steinwart & Christmann (2008, Chapter 4) that the integral operator for $k_0$ and $\mu_0$ is defined by

$$T_{k_0} = I \circ S : L_2(X_0, \mu_0) \to L_2(X_0, \mu_0)$$

$$\text{where} \quad S : L_2(X_0, \mu_0) \to \mathcal{C}(X_0), \quad (Sf)(x) = \int k_0(x, y) f(y) d\mu_0(y), \quad f \in L_2(X_0, \mu_0),$$

and $I : \mathcal{C}(X_0) \hookrightarrow L_2(X_0; \mu_0)$ is the inclusion operator. By the spectral theorem, if $T_{k_0}$ is compact, then there is an at most countable orthonormal set $\{\tilde{e}_j\}_{j \in J}$ of $L_2(X_0, \mu_0)$ and $\{\lambda_j\}_{j \in J}$ with $\lambda_1 \geq \lambda_2 \geq \ldots > 0$ such that $T_{k_0} f = \sum_{j \in J} \lambda_j \langle f, \tilde{e}_j \rangle_{L_2(X_0; \mu_0)} \tilde{e}_j$ for all $f \in L_2(X_0, \mu_0)$. It is easy to see that $\varphi_j := \sqrt{\lambda_j} \tilde{e}_j$ is an orthonormal basis of $\mathcal{H}_0$ (Steinwart & Christmann, 2008).

Our proof is built upon the following two assumptions on the base kernel. The first one asserts that fixing $x$, the energy of $k_0(x, \cdot)$ and $\partial^{0,1} k_0(x, \cdot)$ "concentrates" on the leading eigenfunctions.

**Assumption 1.** *Suppose $k_0(x, x) = 1$ and $\partial^{0,1} k_0(x, \cdot) \in \mathcal{H}_0$ for all $x \in X_0$. For all $\varepsilon > 0$, there exists $N_\varepsilon \in \mathbb{N}$ such that the tail energy of $\partial^{0,1} k_0(x, \cdot)$ beyond the $N_\varepsilon$-th eigenpair is less than $\varepsilon$, uniformly for all $x \in X_0$. That is, denoting $\Phi_m := (\varphi_1, \ldots, \varphi_m)$,*

$$N_\varepsilon := \inf_m \left\{ \left\| \partial^{0,1} k_0(x, \cdot) - \Phi_m \Phi_m^\top \partial^{0,1} k_0(x, \cdot) \right\|_{\mathcal{H}_0} < \varepsilon \text{ for all } x \in X_0 \text{ and} \right.$$

$$\left. \left\| k_0(x, \cdot) - \Phi_m \Phi_m^\top k_0(x, \cdot) \right\|_{\mathcal{H}_0} < \varepsilon \text{ for all } x \in X_0 \right\} < \infty.$$

The second assumption asserts the smoothness and range of eigenfunctions *in a uniform sense*.

**Assumption 2.** *Under Assumption 1, $\{e_j(x) : j \in N_\varepsilon\}$ is uniformed bounded over $x \in X_0$, and the RKHS inner product of $\partial^{0,1} k_0(x, \cdot)$ with $\{e_j : j \in N_\varepsilon\}$ is also uniformly bounded over $x \in X_0$:*

$$M_\varepsilon := \sup_{x \in X_0} \max_{j \in [N_\varepsilon]} \left| \langle \partial^{0,1} k_0(x, \cdot), e_j \rangle_{\mathcal{H}_0} \right| < \infty, \qquad and \qquad Q_\varepsilon := \sup_{x \in X_0} \max_{j \in [N_\varepsilon]} |e_j(x)| < \infty.$$

**Theorem 3.** *Suppose $k_0$, $X_0$, and $\mu_0$ satisfy Assumptions 1 and 2. Let $\{w_j^s : s \in [n], j \in [d]\}$ be sampled i.i.d. from $\mu_0$. Then for any $f$ whose coordinate-wise Nyström approximation (11) and (12) satisfy $\lambda_{\max}(\tilde{P}_G) \leq L^2$, the Lipschitz condition $\lambda_{\max}(G^\top G) \leq L^2 + \varepsilon$ is met with probability $1 - \delta$, as long as $n \geq \tilde{\Theta}\big(\frac{1}{\varepsilon^2} N_\varepsilon^2 M_\varepsilon^2 Q_\varepsilon^2 \log \frac{dN_\varepsilon}{\delta}\big)$, almost independent of $d$. Here $\tilde{\Theta}$ hides all poly-log terms.*

**Satisfaction of Assumptions.** In Appendix A.4 and A.5, we will show that for periodic kernel and Gaussian kernel, Assumptions 1 and 2 hold true with $\tilde{O}(1)$ values of $N_\varepsilon$, $M_\varepsilon$, and $Q_\varepsilon$. It remains open whether non-product kernels such as inverse kernel also enjoy this polynomial sample complexity. Appendix A.6 suggests that the complexity is *quasi-polynomial* for inverse kernels.

## 5 EXPERIMENTAL RESULTS

We studied the empirical robustness and accuracy of the proposed Lipschitz regularisation technique for adversarial training of kernel methods, under both Gaussian kernel and inverse kernel. Comparison will be made with state-of-the-art defense algorithms under effective attacks.

**Datasets.** We tested on three datasets: MNIST, Fashion-MNIST, and CIFAR10. The number of training/validation/test examples for the three datasets are 54k/6k/10k, 54k/6k/10k, 45k/5k/10k, respectively. Each image in MNIST and Fashion-MNIST is represented as a 784-dimensional feature vector, with each feature/pixel normalised to $[0, 1]$. For CIFAR10, we trained it on a residual network to obtain a 512-dimensional feature embedding, which were subsequently normalised to $[0, 1]$. They were used as the input for training all the competing algorithms and were subject to attack.

**Attacks.** To evaluate the robustness of the trained model, we attacked them on test examples using the random initialized Projected Gradient Descent method with 100 steps (PGD, Madry et al., 2018) under two losses: cross-entropy and C&W loss (Carlini & Wagner, 2017). The perturbation $\delta$ was constrained in an $\ell_2$ or $\ell_\infty$ ball. To evaluate robustness, we scaled the perturbation bound $\delta$ from 0.1 to 0.6 for $\ell_\infty$ norm, and from 1 to 6 for $\ell_2$ norm (when $\delta = 6$, the average magnitude per coordinate is 0.214).

**Algorithms.** We compared four training algorithms. The Parseval network orthonormalises the weight matrices to enforce the Lipschitz constant (Cisse et al., 2017). We used three hidden layers of 1024 units and ReLU activation (Par-ReLU). Also considered is the Parseval network with MaxMin activations (Par-MaxMin), which enjoys much improved robustness (Anil et al., 2019). Both algorithms can be customised for $\ell_2$ or $\ell_\infty$ attacks, and were trained under the corresponding norms. Using multi-class hinge loss, they constitute strong baselines for adversarial learning.

Both Gaussian and inverse kernel machines applied Lipschitz regularisation by randomly and greedily selecting $\{w^s\}$, and they will be referred to as Gauss-Lip and Inverse-Lip, respectively. In practice, Gauss-Lip with the coordinate-wise Nyström approximation ($\lambda_{\max}(\tilde{P}_G)$ from Eq (12)) can approximate $\lambda_{\max}(G^\top G)$ with a much smaller number of sample than if using the holistic approximation as in (10). Furthermore, we found an even more efficient approach. Inside the iterative training algorithm, we used L-BFGS to find the input that yields the steepest gradient under the current solution, and then added it to the set $\{w^s\}$ (which was initialized with 15 random points). Although L-BFGS is only a local solver, this greedy approach empirically reduces the number of samples by an order of magnitude. See the empirical convergence results in Appendix A.9. Its theoretical analysis is left for future investigation. We also applied this greedy approach to Inverse-Lip.

**Extending binary kernel machines to multiclass.** The standard kernel methods learn a discriminant function $f^c := \sum_a \gamma_a^c k(x^a, \cdot)$ for each class $c \in [10]$, based on which a large supply of multiclass classification losses can be applied, e.g., CS (Crammer & Singer, 2001) which was used in our experiment. Since the Lipschitz constant of the mapping from $\{f^c\}$ to a real-valued loss is typically at most 1, it suffices to bound the Lipschitz constant of $x \mapsto (f^1(x), \ldots, f^{10}(x))^\top$ by

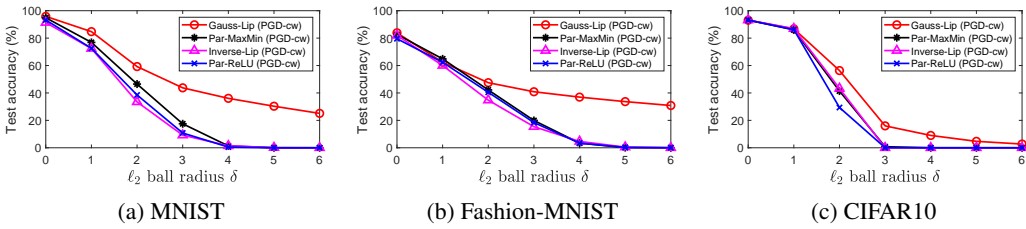

(a) MNIST      (b) Fashion-MNIST      (c) CIFAR10

Figure 3: Test accuracy under PGD attacks on the C&W approximation with $\ell_2$ norm bound

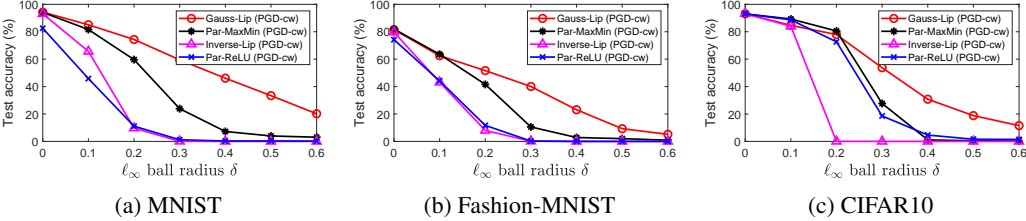

(a) MNIST      (b) Fashion-MNIST      (c) CIFAR10

Figure 4: Test accuracy under PGD attacks on the C&W approximation with $\ell_\infty$ norm bound

$$\max_x \lambda_{\max}(G(x)G(x)^\top) \ \leq \ \max_{\|\varphi\|_{\mathcal{H}}=1} \lambda_{\max}\Big( \sum\nolimits_{c=1}^{10} G_c^\top \varphi\varphi^\top G_c \Big) \ \leq \ L^2, \qquad (14)$$

where $G(x) := [\nabla f^1(x), \cdots, \nabla f^{10}(x)] = [G_1^\top k(x, \cdot), \ldots, G_{10}^\top k(x, \cdot)]$ with $G_c := (g_1^c, \ldots, g_d^c)$. The last term in (14) can be approximated using the same technique as in the binary case. Furthermore, the principle can be extended to $\ell_\infty$ attacks, whose details are relegated to Appendix A.10.

**Parameter selection.** We used the same parameters as in Anil et al. (2019) for training Par-ReLU and Par-MaxMin. To defend against $\ell_2$ attacks, we set $L = 100$ for all algorithms. Gauss-Lip achieved high accuracy and robustness on the validation set with bandwidth $\sigma = 1.5$ for FashionMNIST and CIFAR-10, and $\sigma = 2$ for MNIST. To defend against $\ell_\infty$ attacks, we set $L = 1000$ for all the four methods as in Anil et al. (2019). The best $\sigma$ for Gauss-Lip is 1 for all datasets. Inverse-Lip used 5 stacked layers.

**Results.** Figures 3 and 4 show how the test accuracy decays as an increasing amount of perturbation ($\delta$) in $\ell_2$ and $\ell_\infty$ norm is added to the test images, respectively. Clearly Gauss-Lip achieves higher accuracy and robustness than Par-ReLU and Par-MaxMin on the three datasets, under both $\ell_2$ and $\ell_\infty$ bounded PGD attacks with C&W loss. In contrast, Inverse-Lip only performs similarly to Par-ReLU. Interestingly, we noticed that $\ell_2$ based Par-MaxMin are only slightly better than Par-ReLU under $\ell_2$ attacks, although the former does perform significantly better under $\ell_\infty$ attacks.

For the sake of space, the results for cross-entropy PGD attacks are deferred to Figures 8 and 9 in Appendix A.11. Here cross-entropy PGD attackers find stronger attacks to Parseval networks but not to our kernel models. Our Gauss-Lip again significantly outperforms Par-MaxMin on all the three datasets and under both $\ell_2$ and $\ell_\infty$ norms. The improved robustness of Gauss-Lip does not seem to be attributed to the obfuscated gradient (Athalye et al., 2018), because as shown Figures 3, 4, 8, 9, increased distortion bound does increase attack success, and unbounded attacks drive the success rate to very low. In practice, we also observed that random sampling finds much weaker attacks, and taking 10 steps of PGD is much stronger than just one step.

**Visualization.** The gradient with respect to inputs is plotted in Figure 10 (in the appendix on p. 31) for $\ell_2$ trained Par-MaxMin and Gauss-Lip. The $i$-th row and $j$-th column corresponds to the targeted attack of turning the original class $j$ into a new class $i$, hence the gradient is on the cross-entropy loss with class $i$ as the ground truth. These two figures also explained why Gauss-Lip is more robust than Par-MaxMin: the attacker can easily reduce the targeted cross-entropy loss by following the gradient as shown in Figure 10a, and hence successfully attack Par-MaxMin. In contrast, the gradient shown in Figure 10b does not provide much information on how to flip the class.

**Conclusion.** In this paper, we derived a new certificate for distributional robust risk minimization by using Lipschitz regularization. Application to adversarial learning based on kernel methods exhibited superior robustness, with provably polynomial sample complexity for product kernels. We will apply this function space to GANs to witness the difference between probability distributions, leading to a more stable training scheme as the inner level optimization becomes convex.

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

# Appendix

The pseudo-code of training binary SVMs by enforcing Lipschitz constant is given in Algorithm 1.

---

**Algorithm 1:** Training binary SVMs by enforcing Lipschitz constant $L$

---

1   Initialise the constraint set $S$ by some random samples from $X$.
2   **for** $i = 1, 2, \ldots$ **do**
3   $\quad$ Train SVM using one of the following constraints:

$\qquad$ ① **Brute-force:** $\|\nabla f(w)\|_2^2 \leq L^2, \ \forall \ w \in S$

$\qquad$ ② **Nyström holistic:** $\lambda_{\max}(\tilde{G}^\top \tilde{G}) \leq L^2$ using $S$ as the set $\{w^1, \ldots, w^n\}$ in Eq (10)

$\qquad$ ③ **Nyström coordinate wise:** $\lambda_{\max}(\tilde{P}_G) \leq L^2$ using $S$ as the set $\{w^1, \ldots, w^n\}$ in Eq (12)

4   $\quad$ Let the trained SVM be $f^{(i)}$.
5   $\quad$ Find a new $w$ to add to $S$ by one of the following methods:

$\qquad$ ⓐ **Random:** randomly sample $w$ from $X$.

$\qquad$ ⓑ **Greedy:** find $\operatorname{argmax}_{x \in X} \|\nabla f^{(i)}(x)\|$ (local optimisation) by L-BFGS with 10 random initialisations. Add the distinct results upon convergence to $S$.

6   $\quad$ **Return** if $L^{(i)} := \max_{x \in X} \|\nabla f^{(i)}(x)\|$ falls below $L$.

---

Finding the exact $\operatorname{argmax}_{x \in X} \|\nabla f^{(i)}(x)\|$ is intractable, so we used a local maximum found by L-BFGS with 10 random initialisations as the Lipschitz constant of the current solution $f^{(i)}$ ($L^{(i)}$ in step 6). The solution found by L-BFGS is also used as the new greedy point added in step 5b.

Furthermore, the kernel expansion $f(x) = \frac{1}{l} \sum_{a=1}^l \gamma_a k(x^a, \cdot)$ can lead to high cost in optimisation (our experiment used $l = 54000$), and therefore we used *another* Nyström approximation for the kernels. We randomly sampled 1000 landmark points, and based on them we computed the Nyström approximation for each $k(x^a, \cdot)$, denoted as $\tilde{\varphi}(x^a) \in \mathbb{R}^{1000}$. Then $f(x)$ can be written as $\frac{1}{l} \sum_{a=1}^l \gamma_a \tilde{\varphi}(x^a)^\top \tilde{\varphi}(x)$. Defining $w = \frac{1}{l} \sum_{a=1}^l \gamma_a \tilde{\varphi}(x^a)$, we can equivalently optimise over $w$, and the RKHS norm bound on $f$ can be equivalently imposed as the $\ell_2$-norm bound on $w$.

To summarise, Nyström approximation is used in two different places: one for approximating the kernel function, and one for computing $\|g_j\|_{\mathcal{H}}$ either holistically or coordinate wise. For the former, we randomly sampled 1000 landmark points; for the latter, we used greedy selection as option b in step 5 of Algorithm 1.

**Detailed algorithm for multiclass classification.** It is easy to extend Algorithm 1 to multiclass. For example, with MNIST dataset, we solve the following optimisation problem to defend $\ell_2$ attacks:

$$\min_{\gamma^1, \ldots, \gamma^{10}} \quad \sum_{i=1}^n \ell(F(x), \boldsymbol{y}), \quad \text{where } F = \left[ \sum_{i=1}^n \boldsymbol{\gamma}_i^1 k(x_i, \cdot); \ldots; \sum_{i=1}^n \boldsymbol{\alpha}_i^{10} k(x_i, \cdot) \right]$$

$$s.t. \quad \sup_{\|\varphi\|_{\mathcal{H}} \leq 1} \lambda_{\max}\left( \sum_{c=1}^{10} G_c^\top \varphi \varphi^\top G_c \right) \approx \boxed{\sup_{\|v\|_2 \leq 1} \lambda_{\max}\left( \sum_{c=1}^{10} \tilde{G}_c^\top v v^\top \tilde{G}_c \right) \leq L^2,}$$

where $\ell(F(x), \boldsymbol{y})$ is the Crammer & Singer loss, and the constraint is derived from (14) by using its Nyström approximation $\tilde{G}_c = [\tilde{g}_1^c, \ldots, \tilde{g}_d^c]$, which depends on $\{\gamma^1, \ldots, \gamma^{10}\}$ linearly. Note that the constraint itself is a supremum problem:

$$\sup_{\|v\|_2 \leq 1} \lambda_{\max}\left( \sum_{c=1}^{10} \tilde{G}_c^\top v v^\top \tilde{G}_c \right) = \sup_{\|v\|_2 \leq 1, \|u\|_2 \leq 1} u^\top \left( \sum_{c=1}^{10} \tilde{G}_c^\top v v^\top \tilde{G}_c \right) u.$$

Since there is only one constraint, interior point algorithm is efficient. It requires the gradient of the constraint, which can be computed by Danskin's theorem. In particular, we alternates between

updating $v$ and $u$, until they converge to the optimal $v_*$ and $u_*$. Finally, the derivative of the constraint with respect to $\{\boldsymbol{\gamma}^c\}$ can be calculated from $\sum_{c=1}^{10}(u_*^\top \tilde{G}_c^\top v_*)^2$, as a function of $\{\boldsymbol{\gamma}^c\}$.

To defend $\ell_\infty$ attacks, we need to enforce the $\ell_\infty$ norm of the Jacobian matrix:

$$
\begin{aligned}
\sup_{x \in X} \left\| \left[ g^1(x), \ldots, g^{10}(x) \right]^\top \right\|_\infty &= \sup_{x \in X} \max_{1 \le c \le 10} \| g^c(x) \|_1 \\
&= \max_{1 \le c \le 10} \sup_{x \in X} \| g^c(x) \|_1 \\
&\le \max_{1 \le c \le 10} \sup_{\|\varphi\|_2 \le 1, \|u\|_\infty \le 1} u^\top \tilde{G}_c^\top \varphi,
\end{aligned}
$$

where the last inequality is due to

$$
\sup_{x \in X} \| g(x) \|_1 = \sup_{x \in X} \sup_{\|u\|_\infty \le 1} u^\top g(x) \le \sup_{\|v\|_2 \le 1, \|u\|_\infty \le 1} u^\top \tilde{G}^\top v.
$$

Therefore, the overall optimisation problem to defense $\ell_\infty$ attacks is

$$
\min_{\boldsymbol{\gamma}^1, \ldots, \boldsymbol{\gamma}^{10}} \sum_{i=1}^n \ell(F(x), \boldsymbol{y}), \quad \text{where } F = \left[ \sum_{i=1}^n \boldsymbol{\gamma}_i^1 k(x_i, \cdot); \ldots; \sum_{i=1}^n \boldsymbol{\gamma}_i^{10} k(x_i, \cdot) \right]
$$

$$
s.t. \quad \sup_{\|v\|_2 \le 1, \|u\|_\infty \le 1} u^\top \tilde{G}_c^\top v \le L, \qquad \forall c \in \{1, \ldots, 10\} \tag{15}
$$

For each $c$, we alternatively update $v$ and $u$ in (15), converging to the optimal $v_*$ and $u_*$. Finally, the derivative of $\sup_{\|v\|_2 \le 1, \|u\|_\infty \le 1} u^\top \tilde{G}_c^\top v$ with respect to $\boldsymbol{\gamma}^c$ can be calculated from $u_*^\top \tilde{G}_c^\top v_*$, as a function of $\boldsymbol{\gamma}^c$.

## A  PROOFS OF RESULTS

The following appendix contains the complete set of proofs and auxiliary results.

### PROOFS FOR §3: CERTIFICATE FOR DISTRIBUTIONAL ROBUSTNESS

Duality results like Lemma 1 have been the basis of a number of recent theoretical efforts in the theory of adversarial learning (Blanchet et al., 2016; Gao & Kleywegt, 2016; Shafieezadeh-Abadeh et al., 2017; Sinha et al., 2018), the results of Blanchet & Murthy (2019) being the most general to date.

**Lemma 1 (Blanchet & Murthy (2019, Thm. 1)).** *Assume $\Omega$ is a Polish space and fix $\mu \in \mathfrak{P}(\Omega)$. Let $c : \Omega \times \Omega \to \bar{\mathbb{R}}_{\ge 0}$ be lower semicontinuous with $c(\omega, \omega) = 0$ for all $\omega \in \Omega$, and $f \in \mathscr{L}_1(\Omega, \mu)$ is upper semicontinuous. Then for all $r \ge 0$ there is*

$$
\sup_{\nu \in B_c(\mu, r)} \int f \, d\nu = \inf_{\lambda \ge 0} \left( \lambda r + \int f^{\lambda c} \, d\mu \right). \tag{16}
$$

The necessity for such duality results like Lemma 1 is because while the supremum on the left hand side of (16) is over a (usually) infinite dimensional space, the right hand side only involves only a finite dimensional optimisation. The generalised conjugate in (16) also hides an optimisation, but when the outcome space $\Omega$ is finite dimensional, this too is a finite dimensional problem.

The following is sometimes stated a consequence of or in the proof of the McShane–Whitney extension theorem, but it is immediate to observe.

**Lemma 2 (McShane–Whitney).** *Let $X$ be a set. Assume $c : X \times X \to \bar{\mathbb{R}}_{\ge 0}$ satisfies $c(x, x) = 0$ for all $x \in X$, and $f : X \to \mathbb{R}$. Then*

$$
\forall x, y \in X : \ f(x) - f(y) \le \lambda c(x, y) \implies \forall y \in X : \ f(y) = \sup_{x \in X} \big( f(x) - \lambda c(x, y) \big).
$$

**Lemma 3.** *Assume $X$ is a locally convex Hausdorff topological vector space. Let $c : X \to \bar{\mathbb{R}}$ be lower semicontinuous, sublinear, and continuous at $0$, let $f : X \to \bar{\mathbb{R}}$ be closed convex. Then for $\lambda > 0$ there is*

$$
\forall y \in X : \ \sup_{x \in X} \big( f(x) - \lambda c(x - y) \big) = \begin{cases} f(y) & \partial f(X) \subseteq \partial \lambda c(0) \\ \infty & \partial f(X) \not\subseteq \partial \lambda c(0). \end{cases}
$$

*Proof.* Because $f$ is closed convex, it is equal to its biconjugate (Zălinescu, 2002, Thm. 2.3.3), because $c$ is sublinear and lower semicontinuous $\lambda c(x) = \sup_{x^* \in \partial \lambda c(0)} \langle x, x^* \rangle$ for all $x \in X$ (Zălinescu, 2002, Thm. 2.4.14 (iv)). It follows that

$$\sup_{x \in X} \left( f(x) - \lambda c(x-y) \right) = \sup_{x \in X} \sup_{x^* \in \partial f(X)} \inf_{g^* \in \partial \lambda c(0)} \left( \langle x, x^* \rangle - f^*(x^*) - \langle g^*, x-y \rangle \right)$$

$$= \sup_{x \in X} \sup_{x^* \in \partial f(X)} \inf_{g^* \in \partial \lambda c(0)} \left( \langle x, x^* - g^* \rangle + \langle y, g^* \rangle - f^*(x^*) \right).$$

Because $c$ is continuous at 0, $\partial c(0)$ is weak$^*$-compact and convex (Zălinescu, 2002, Thm. 2.4.9), and so we can apply a minimax theorem (Zălinescu, 2002, Thm. 2.10.2) to produce

$$\sup_{x \in X} \sup_{x^* \in \partial f(X)} \inf_{g^* \in \partial \lambda c(0)} \left( \langle x, x^* - g^* \rangle + \langle y, g^* \rangle - f^*(x^*) \right)$$

$$= \sup_{x^* \in \partial f(X)} \inf_{g^* \in \partial \lambda c(0)} \sup_{x \in X} \left( \langle x, x^* - g^* \rangle + \langle y, g^* \rangle - f^*(x^*) \right)$$

$$= \sup_{x^* \in \partial f(X)} \inf_{g^* \in \partial \lambda c(0)} \begin{cases} \langle y, x^* \rangle - f^*(x^*) & g^* = x^* \\ \infty & g^* \neq x^* \end{cases}$$

$$= \sup_{x^* \in \partial f(X)} \begin{cases} \langle y, x^* \rangle - f^*(x^*) & x^* \in \partial \lambda c(0) \\ \infty & x^* \notin \partial \lambda c(0) \end{cases}$$

$$= \begin{cases} f(y) & \partial f(X) \subseteq \partial \lambda c(0) \\ \infty & \partial f(X) \nsubseteq \partial \lambda c(0), \end{cases}$$

as claimed. ■

*Remark* 1. The minimisation of $g - h$, where $g$ and $h$ are convex functions, is called difference convex (DC) programming (Hiriart-Urruty, 1989). The condition in Lemma 3 bears a striking resemblance to the common necessary condition (e.g. Hiriart-Urruty, 1989; Penot, 1998) for such problems

$$x \in \operatorname*{arginf}_{x' \in X} f(x') \implies \partial h(x) \subseteq \partial g(x).$$

Likewise there are similar sufficient conditions. The proof of Lemma 4 is also quite similar to the proofs of the Toland (1979) duality formula (viz. Hiriart-Urruty, 1986)

$$\inf_{x \in X} (g(x) - h(x)) = \inf_{x^* \in X^*} (h^*(x^*) - g^*(x^*)),$$

which suggests that the principles of Lemma 4 may be more general. A generalisation to a general convex function $c$ satisfying $c(0) = 0$, would remove the positive homogeneity requirement of Lemma 4, and allow any translation invariant metric in place of $c$. The assumptions we have made are compatible with metrics which arise from norms, that is, the translation invariant and positively homogeneous metrics.

**Lemma 4.** *Assume $X$ is a topological vector space. Let $c : X \to \bar{\mathbb{R}}_{\geq 0}$, and $f : X \to \mathbb{R}$. Then for $\lambda > 0$ there is*

$$\forall x, y \in X : \ f(x) - f(y) \leq \lambda c(x - y) \iff \partial f(X) \subseteq \partial \lambda c(0).$$

*Proof.* Suppose $\lambda > 0$ is such that $f(x) - f(y) \leq \lambda c(x-y)$ for all $x, y \in X$. Let $x^* \in \partial f(X)$. Then there is $x \in X$ with

$$\forall y \in X : \ \langle y - x, x^* \rangle \leq f(y) - f(x) \leq \lambda c(y-x)$$
$$\implies \forall y \in X : \ \langle y, x^* \rangle \leq f(y+x) - f(x) \leq \lambda c(y),$$

this shows $x^* \in \partial \lambda c(0)$. Next assume $\lambda > 0$ satisfies $\partial f(X) \subseteq \partial \lambda c(0)$. Then

$$\forall x \in X, \exists x^* \in \partial f(x), \forall y \in X : \ f(x) - f(y) \leq \langle x - y, x^* \rangle \leq \lambda c(x - y),$$

where the second inequality is because $x^* \in \partial \lambda c(0)$ for all $x^* \in \partial f(x)$. ■

**Theorem 1.** *Assume $X$ is a separable Fréchet space and fix $\mu \in \mathfrak{P}(X)$. Assume $c : X \to \bar{\mathbb{R}}_{\geq 0}$ is sublinear and continuous, and $f \in \mathscr{L}_1(X, \mu)$ is upper semicontinuous. Then for all $r \geq 0$,*

$$DRR := \sup_{\nu \in \mathrm{B}_c(\mu, r)} \mathrm{risk}_\ell(f, \nu) \leq r \, \mathrm{lip}_c(\ell_f) + \mathrm{risk}_\ell(f, \mu).$$

*The **tightness** of the bound can be quantified as follows. Let $\Delta(\mu) := r \operatorname{lip}_c(\ell_f) + \operatorname{risk}_\ell(f, \mu) - \sup_{\nu \in \mathrm{B}_c(\mu, r)} \operatorname{risk}_\ell(f, \nu)$. If $\operatorname{lip}_c(f) < \infty$ then*

$$\Delta(\mu) \leq r \left( \operatorname{lip}_c(\ell_f) - \left[ \operatorname{lip}_c(\overline{\mathrm{co}}\,\ell_f) - \frac{1}{r} \int (\ell_f - \overline{\mathrm{co}}\,\ell_f)\,\mathrm{d}\mu \right]_+ \right),$$

*where $[\,\cdot\,]_+ := \max\{\,\cdot\,, 0\}$ and $1/0 := \infty$, so that when $\ell_f$ is closed convex there is **equality** in* (3).

*Proof.* Because $c$ is assumed sublinear, it is positively homogeneous and there is $c(x, x) = c(x - x) = c(0) = 0$ for all $x \in X$. Therefore we can apply Lemma 1 and Lemma 2 to obtain

$$\sup_{\nu \in \mathrm{B}_c(\mu, r)} \int \ell_f\,\mathrm{d}\nu = \inf_{\lambda \geq 0} \left[ r\lambda + \int \ell_f^{\lambda c}\,\mathrm{d}\mu \right]$$

$$\leq \inf_{\lambda \geq \operatorname{lip}_c(\ell_f)} \left[ r\lambda + \int \ell_f^{\lambda c}\,\mathrm{d}\mu \right]$$

$$= r \operatorname{lip}_c(\ell_f) + \int \ell_f\,\mathrm{d}\mu.$$

Observing that $\overline{\mathrm{co}}\,\ell_f \leq \ell_f$, applying Lemma 3 and Lemma 4 we find for all $x \in X$

$$\sup_{\lambda \in [0, \infty)} \left( \ell_f(x) - \ell_f^{\lambda c}(x) - r\lambda \right) = \sup_{\lambda \in [0, \infty)} \left( \ell_f(x) - \sup_{y \in X} \left( \ell_f(y) - \lambda c(x - y) \right) - r\lambda \right)$$

$$= \sup_{\lambda \in [0, \infty)} \inf_{y \in X} \left( \ell_f(x) - \ell_f(y) + \lambda c(x - y) - r\lambda \right)$$

$$\leq \sup_{\lambda \in [0, \infty)} \inf_{y \in X} \left( \ell_f(x) - \overline{\mathrm{co}}\,\ell_f(y) + \lambda c(x - y) - \lambda r \right)$$

$$= \sup_{\lambda \in [0, \infty)} \left( \ell_f(x) - \overline{\mathrm{co}}\,\ell_f(x) - \infty \left[\!\left[ \operatorname{lip}_c(\overline{\mathrm{co}}\,\ell_f) > \lambda \right]\!\right] - \lambda r \right)$$

$$= \ell_f(x) - \overline{\mathrm{co}}\,\ell_f(x) - r \operatorname{lip}_c(\overline{\mathrm{co}}\,\ell_f). \tag{1}$$

Similarly, for all $x \in X$ there is

$$\sup_{\lambda \in [0, \infty)} \left( \ell_f(x) - \ell_f^{\lambda c}(x) - r\lambda \right) \leq \sup_{\lambda \in [0, \infty)} \left( \ell_f(x) - \ell_f^{\lambda c}(x) \right) + \sup_{\lambda \in [0, \infty)} -r\lambda$$

$$= \sup_{\lambda \in [0, \infty)} \left( \ell_f(x) - \ell_f^{\lambda c}(x) \right)$$

$$= \sup_{\lambda \in [0, \infty)} \inf_{y \in X} \left( \ell_f(x) - \ell_f(y) + \lambda c(x - y) \right)$$

$$\leq \inf_{y \in X} \sup_{\lambda \in [0, \infty)} \left( \ell_f(x) - \ell_f(y) + \lambda c(x - y) \right)$$

$$= \inf_{y \in X} \begin{cases} \infty & c(x - y) > 0 \\ 0 & c(x - y) = 0 \end{cases}$$

$$= 0. \tag{2}$$

Then, using (1) and (2) we find

$$\left( r \operatorname{lip}_c(\ell_f) + \int \ell_f\,\mathrm{d}\mu \right) - \inf_{\lambda \in [0, \infty)} \left( r\lambda - \int \ell_f^{\lambda c}\,\mathrm{d}\mu \right)$$

$$= r \operatorname{lip}_c(\ell_f) + \sup_{\lambda \in [0, \infty)} \int \left( \ell_f - \ell_f^{\lambda c} - \lambda r \right) \mathrm{d}\mu$$

$$\leq r \operatorname{lip}_c(\ell_f) + \int \sup_{\lambda \in [0, \infty)} \left( \ell_f - \ell_f^{\lambda c} - \lambda r \right) \mathrm{d}\mu$$

$$\overset{(1),(2)}{\leq} r \operatorname{lip}_c(\ell_f) + \min \left\{ \int (\ell_f - \overline{\mathrm{co}}\,\ell_f)\,\mathrm{d}\mu - r \operatorname{lip}_c(\overline{\mathrm{co}}\,\ell_f), 0 \right\}.$$

The proof is complete. ∎

**Proposition 1.** *Assume $X$ is a separable Fréchet space with $X_0 \subseteq X$. Assume $c : X \to \bar{\mathbb{R}}_{\geq 0}$ is sublinear and continuous, and $\ell_f \in \bigcap_{\mu \in \mathfrak{P}(X_0)} \mathscr{L}_1(X, \mu)$ is upper semicontinuous, has $\mathrm{lip}_c(\ell_f) < \infty$, and attains its maximum on $X_0$. Then for all $r \geq 0$ with $1/0 := \infty$,*

$$\sup_{\mu \in \mathfrak{P}(X_0)} \Delta(\mu) = r\left( \mathrm{lip}_c(\ell_f) - \left[ \mathrm{lip}_c(\overline{\mathrm{co}}\, \ell_f) - \frac{1}{r}\, \rho(\ell_f) \right]_+ \right).$$

*Proof.* Let $x_0 \in X_0$ be be the point at which $\ell_f(x_0) = \sup_{x \in X_0} \ell_f(x)$. Then

$$\Delta(\delta_{x_0}) = r\,\mathrm{lip}_c(\ell_f) + \int \ell_f \, \mathrm{d}\delta_{x_0} - \sup_{\nu \in \mathrm{B}_c(\delta_{x_0}, r)} \int \ell_f \, \mathrm{d}\delta_{x_0}$$

$$= r\,\mathrm{lip}_c(\ell_f) + \int \ell_f \, \mathrm{d}\delta_{x_0} - \int \ell_f \, \mathrm{d}\delta_{x_0}$$

$$= r\,\mathrm{lip}_c(\ell_f). \tag{3}$$

Then there is

$$r\,\mathrm{lip}_c(\ell_f) \overset{(3)}{\leq} \sup_{\mu \in \mathfrak{P}(X_0)} \Delta(\mu) \overset{(4)}{\leq} r\left( \mathrm{lip}_c(\ell_f) - \max\left\{ \mathrm{lip}_c(\overline{\mathrm{co}}\, \ell_f) - \frac{1}{r}\, \rho(\ell_f), 0 \right\} \right) \leq r\,\mathrm{lip}_c(\ell_f),$$

which completes the proof. ∎

*Remark* 2. When $f : \mathbb{R}^n \to \bar{\mathbb{R}}$ satisfies $f \not\equiv \infty$ and $f$ is minorised by an affine function, there is (cf. Hiriart-Urruty & Lemaréchal, 2010, Prop. 1.5.4)

$$\forall x \in \mathbb{R}^n : \ \overline{\mathrm{co}}\, f(x) = \inf\left\{ \sum_{i \in [n+1]} \alpha_i f(x_i) \,\Big|\, \sum_{i \in [n+1]} \alpha_i = 1, \ x = \sum_{i \in [n+1]} \alpha_i x_i \right\},$$

where the infimum is over all sequences $(\alpha_i)_{i \in [n+1]}$ and $(x_i)_{i \in [n+1]} \subseteq \mathbb{R}^n$ satisfying the conditions above. Consequentially there is the common expression

$$\rho(f) = \sup\left\{ f\left( \sum_{i \in [n+1]} \alpha_i x_i \right) - \sum_{i \in [n+1]} \alpha_i f(x_i) \,\Big|\, (\alpha_i, x_i)_{i \in [n+1]} \subseteq \mathbb{R}_{\geq 0} \times \mathbb{R}^n, \ \sum_{i \in [n+1]} \alpha_i = 1 \right\}.$$

In Lemma 5, by the weak* topology on $\mathfrak{P}(\Omega)$ we mean the coarsest topolgoy on $\mathfrak{P}(\Omega)$ that makes the bounded continuous functions on $\Omega$ its topological dual space. Likewise $\rightharpoonup^*$ denotes convergence in this topology.

**Lemma 5.** *Assume $(\Omega, c)$ is a compact Polish space and $\mu \in \mathfrak{P}(\Omega)$ is non-atomic. For any $\nu^\star \in \mathfrak{P}(\Omega)$ and $r > 0$ there is a sequence $(f_i)_{i \in \mathbb{N}} \subseteq A_\mu(r) := \{f \in \mathscr{L}_0(\Omega, \Omega) \mid \int c\,\mathrm{d}(\mathrm{id}, f)_{\#}\mu \leq r\}$ with $(f_i)_{\#}\mu \rightharpoonup^* \nu^\star$.*

*Proof.* Let $P(\mu, \nu) := \{f \in \mathscr{L}_0(X, X) \mid f_{\#}\mu = \nu\}$. Since $\mu$ is non-atomic and $c$ is continuous Pratelli (2007, Thm. B) shows

$$\forall \nu \in \mathfrak{P}(\Omega) : \ \inf_{f \in P(\mu, \nu)} \int c\,\mathrm{d}(\mathrm{id}, f)_{\#}\mu = \mathrm{cost}_c(\mu, \nu).$$

Let $r^\star := \mathrm{cost}_c(\mu, \nu^\star)$, obviously $r^\star \leq r$. Assume $r^\star > 0$, otherwise the lemma is trivial. Fix a sequence $(\varepsilon_k)_{k \in \mathbb{N}} \subseteq (0, r^\star)$ with $\varepsilon_k \to 0$. For $u \geq 0$ let $\nu(u) := \mu + u(\nu^\star - \mu)$. Then

$$\mathrm{cost}_c(\mu, \nu(0)) = 0 \ \text{and} \ \mathrm{cost}_c(\mu, \nu(1)) = r^\star,$$

and because $\mathrm{cost}_c$ metrises the weak* topology on $\mathfrak{P}(\Omega)$ (Villani, 2008, Thm. 6.9), the mapping $u \mapsto \mathrm{cost}_c(\mu, \nu(u))$ is continuous. Then by the intermediate value theorem for every $k \in \mathbb{N}$ there is some $u_k > 0$ with $\mathrm{cost}_c(\mu, \nu(u_k)) = r^\star - \varepsilon_k$, forming a sequence $(u_k)_{k \in \mathbb{N}} \subseteq [0, 1]$. Then for every $k$ there is a sequence $(f_{jk})_{j \in \mathbb{N}} \subseteq P(\mu, \nu(u_k))$ so that $(f_{jk})_{\#}\mu \rightharpoonup *\nu(k)$ and

$$\lim_{j \in \mathbb{N}} \int c\,\mathrm{d}(\mathrm{id}, f_{jk})_{\#}\mu = \inf_{f \in P(\mu, \nu(k))} \int c\,\mathrm{d}(\mathrm{id}, f_k)_{\#}\mu = \mathrm{cost}_c(\mu, \nu(k)) = r^\star - \varepsilon_k.$$

Therefore for every $k \in \mathbb{N}$ there exists $j_k \geq 0$ so that for every $j \geq j_k$

$$\int c \, \mathrm{d}(\mathrm{id}, f_{jk})_{\#} \mu \leq r^{\star}. \tag{3}$$

Let us pass directly to this subsequence of $(f_{jk})_{j \in \mathbb{N}}$ for every $k \in \mathbb{N}$ so that (3) holds for all $j, k \in \mathbb{N}$. Next by construction we have $\nu(u_k) \to \nu^{\star}$. Therefore $(f_{jk})_{j,k \in N}$ has a subsequence in $k$ so that $(f_{jk})_{\#} \mu \rightharpoonup^{*} \nu^{\star}$. By ensuring (3) is satisfied, the sequences $(f_{jk})_{j \in \mathbb{N}} \subseteq A_{\mu}(r)$ for every $k \in \mathbb{N}$. ∎

**Theorem 2.** *Assume $(X, c)$ is a separable Banach space. Fix $\mu \in \mathfrak{P}(X)$ and let $R_{\mu}(r) := \{g \in \mathcal{L}_0(X, \mathbb{R}_{\geq 0}) \mid \int g \, \mathrm{d}\mu \leq r\}$. Then for $f \in \mathcal{L}_0(\Omega, \bar{\mathbb{R}})$, $r > 0$ there is*

$$\text{variable-radius risk} := \sup_{g \in R_{\mu}(r)} \int \mu(\mathrm{d}\omega) \sup_{\omega' \in \mathrm{B}_c(\omega, g(\omega))} \ell_f(\omega') \leq \sup_{\nu \in \mathrm{B}_c(\mu, r)} \text{risk}_{\ell}(f, \nu) = DRR.$$

*The equality holds in (6) if $\mu$ is non-atomically concentrated on a compact subset of $X$, on which $f$ is continuous with the subspace topology.*

*Proof.* Inequality (6). For $g \in R_{\mu}(r)$, let $\Gamma_g : X \rightrightarrows X$ denote the set-valued mapping with $\Gamma_g(x) := \mathrm{B}_c(x, g(x))$. Let $\mathcal{L}_0(X, \Gamma_g)$ denote the set of Borel $a : X \to X$ so that $a(x) \in \Gamma_g(x)$ for $\mu$-almost all $x \in X$. Let $A_{\mu}(r) := \bigcup_{g \in \mathbb{R}_{\mu}(r)} \mathcal{L}_0(X, \Gamma_g)$. Clearly for every $a \in A_{\mu}(r)$ there is

$$r \geq \int c(x, a(x)) \, \mathrm{d}\mu = \int c \, \mathrm{d}(\mathrm{id}, a)_{\#} \mu,$$

which shows $\{a_{\#}\mu \mid a \in A_{\mu}(r)\} \subseteq \mathrm{B}_c(\mu, r)$. Then if there is equality in (4), we have

$$\sup_{g \in R_{\mu}(r)} \int \sup_{x' \in \Gamma_g(x)} f(x) = \sup_{g \in R_{\mu}(r)} \sup_{a \in \mathcal{L}_0(X, \Gamma_g)} \int f \, \mathrm{d}a_{\#}\mu \tag{4}$$

$$= \sup_{a \in A_{\mu}(r)} \int f \, \mathrm{d}a_{\#}\mu$$

$$\leq \sup_{\nu \in \mathrm{B}_c(\mu, r)} \int f \, \mathrm{d}a_{\#}\nu,$$

which proves the inequality (6). □

*Equality* (4). To complete the proof we will now justify the exchange of integration and supremum in (4). The set $\mathcal{L}_0(X, \Gamma_g)$ is trivially decomposable (Giner, 2009, see the remark at the bottom of p. 323, Def. 2.1). By assumption $f$ is Borel measurable. Since $f$ is measurable, any decomposable subset of $\mathcal{L}_0(X, X)$ is $f$-decomposable (Giner, 2009, Prop. 5.3) and $f$-linked (Giner, 2009, Prop. 3.7 (i)). Giner (2009, Thm. 6.1 (c)) therefore allows us to exchange integration and supremum in (4). □

*Equality in* (6). Under the additional assumptions there exists $\nu^{\star} \in \mathfrak{P}(\Omega)$ with (via Blanchet & Murthy, 2019, Prop. 2)

$$\int f \, \mathrm{d}\nu^{\star} = \sup_{\nu \in \mathrm{B}_c(\mu, r)} \int f \, \mathrm{d}\nu.$$

The compact subset where $\mu$ is concentrated and non-atomic is a Polish space with the Banach metric. Therefore Using Lemma 5 there is a sequence $(f_i)_{i \in \mathbb{N}} \subseteq A_{\mu}(r)$ so that

$$\lim_{i \in \mathbb{N}} \int f_i \, \mathrm{d}\mu = \int f \, \mathrm{d}\nu^{\star} = \sup_{\nu \in \mathrm{B}_c(\mu, r)} \int f \, \mathrm{d}\nu,$$

proving equality in (6). □

∎

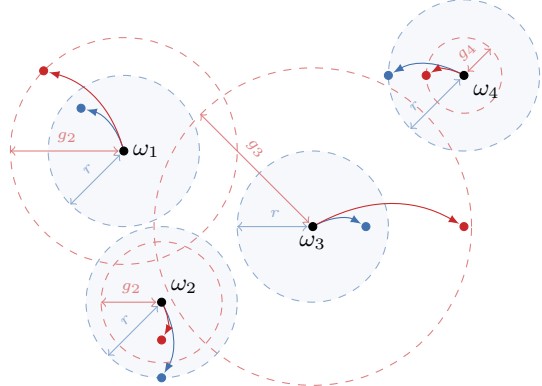

Figure 5: In the classical adversarial risk (5) the perturbation size at each point is at most $r$ (blue), however with the variable-radius risk the *expected* perturbation size is at most $r$.

PROOFS FOR §4: PROVABLE LIPSCHITZ REGULARISATION FOR KERNEL METHODS

**Theorem 3.** *Suppose $k_0$, $X_0$, and $\nu_0$ satisfy Assumptions 1 and 2. Let $\{w_j^s : s \in [n], j \in [d]\}$ be sampled i.i.d. from $\nu_0$. Then for any $f$ whose coordinate-wise Nyström approximation* (11) *and* (12) *satisfy $\lambda_{\max}(\tilde{P}_G) \leq L^2$, the Lipschitz condition $\lambda_{\max}(G^\top G) \leq L^2 + \varepsilon$ is met with probability $1 - \delta$, as long as $n \geq \tilde{\Theta}\big(\frac{1}{\varepsilon^2} N_\varepsilon^2 M_\varepsilon^2 Q_\varepsilon^2 \log \frac{dN_\varepsilon}{\delta}\big)$, almost independent of $d$. Here $\tilde{\Theta}$ hides all poly-log terms.*

PROOFS AND MORE RESULTS FOR §4: KERNEL APPROXIMATION

A.1 RANDOM SAMPLING REQUIRES EXPONENTIAL COST

The most natural idea of leveraging the samples is to add the constraints $\|g(w^s)\| \leq L$. For Gaussian kernel, we may sample from $\mathcal{N}(\mathbf{0}, \sigma^2 I)$ while for inverse kernel we may sample uniformly from $B$. This leads to our training objective:

$$\min_{f \in \mathcal{H}} \quad \frac{1}{l} \sum_{i=1}^{l} \text{loss}(f(x^i), y^i) + \frac{\lambda}{2} \|f\|_{\mathcal{H}}^2 \qquad s.t. \qquad \|g(w^s)\| \leq L, \quad \forall s \in [n].$$

Unfortunately, this method may require $O(\frac{1}{\varepsilon^d})$ samples to guarantee $\sum_j \|g_j\|_{\mathcal{H}}^2 \leq L^2 + \varepsilon$ w.h.p. This is illustrated in Figure 6, where $k$ is the polynomial kernel with degree 2 whose domain $X$ is the unit ball $B$, and $f(x) = \frac{1}{2}(v^\top x)^2$. We seek to test whether the gradient $g(x) = (v^\top x)v$ has norm bounded by 1 for all $x \in B$, and we are only allowed to test whether $\|g(w^s)\| \leq 1$ for samples $w^s$ that are drawn uniformly at random from $B$. This is equivalent to testing $\|v\| \leq 1$, and to achieve it at least one $w^s$ must be from the $\varepsilon$ ball around $v/\|v\|$ or $-v/\|v\|$, intersected with $B$. But the probability of hitting such a region decays exponentially with the dimensionality $d$.

The key insight from the above counter-example is that in fact $\|v\|$ can be easily computed by $\sum_{s=1}^{d}(v^\top \tilde{w}_s)^2$, where $\{\tilde{w}^s\}_{s=1}^{d}$ is the *orthonormal* basis computed from the Gram–Schmidt process on $d$ random samples $\{w^s\}_{s=1}^{d}$ ($n = d$). With probability 1, $n$ samples drawn uniformly from $B$ must span $\mathbb{R}^d$ as long as $n \geq d$, i.e., $\text{rank}(W) = d$ where $W = (w^1, \ldots, w^n)$. The Gram–Schmidt process can be effectively represented using a pseudo-inverse matrix (allowing $n > d$) as

$$\|v\|_2 = \left\| (W^\top W)^{-1/2} W^\top v \right\|_2,$$

where $(W^\top W)^{-1/2}$ is the square root of the pseudo-inverse of $W^\top W$. This is exactly the intuition underlying the Nyström approximation that we will leveraged.

A.2 SPECTRUM OF KERNELS

Let $k$ be a continuous kernel on a compact metric space $X$, and $\mu$ be a finite Borel measure on $X$ with $supp[\mu] = X$. We will re-describe the following spectral properties in a more general way than

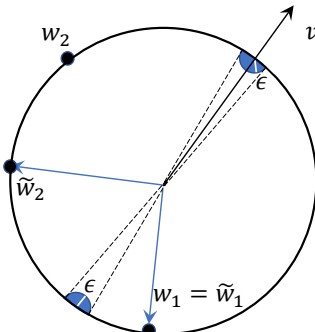

Figure 6: Suppose we use a polynomial kernel with degree 2, and $f(x) = \frac{1}{2}(v^\top x)^2$ for $x \in B$. Then $g(x) = (v^\top x)v$. If we want to test whether $\sup_{x \in B} \|g(x)\|_2 \leq 1$ by evaluating $\|g(w)\|_2$ on $w$ that is randomly sampled from $B$ such as $w_1$ and $w_2$, we must sample within the $\varepsilon$ balls around the intersection of $B$ and the ray along $v$ (both directions). See the blue shaded area. The problem, however, becomes trivial if we use the orthonormal basis $\{\tilde{w}_1, \tilde{w}_2\}$.

in §4. Recall from Chapter 4 of Steinwart & Christmann (2008) that the integral operator for $k$ and $\mu$ is defined by

$$T_k = I_k \circ S_k : L_2(X, \mu) \to L_2(X, \mu)$$

where $S_k : L_2(X; \mu) \to C(X), \quad (S_k f)(x) = \int k(x, y) f(y) d\mu(y), \quad f \in L_2(X, \mu),$

$I_k : C(X) \hookrightarrow L_2(X; \mu),$ inclusion operator.

By the spectral theorem, if $T_k$ is compact, then there is an at most countable orthonormal set (ONS) $\{\tilde{e}_j\}_{j \in J}$ of $L_2(X, \mu)$ and $\{\lambda_j\}_{j \in J}$ with $\lambda_1 \geq \lambda_2 \geq \ldots > 0$ such that

$$Tf = \sum_{j \in J} \lambda_j \langle f, \tilde{e}_j \rangle_{L_2(X; \mu)} \tilde{e}_j, \qquad f \in L_2(X, \mu).$$

In particular, we have $\langle \tilde{e}_i, \tilde{e}_j \rangle_{L_2(X; \mu)} = \delta_{ij}$ (i.e., equals 1 if $i = j$, and 0 otherwise), and $T\tilde{e}_i = \lambda_i \tilde{e}_i$. Since $\tilde{e}_j$ is an equivalent class instead of a single function, we assign a set of continuous functions $e_j = \lambda_j^{-1} S_k \tilde{e}_j \in C(X)$, which clearly satisfies

$$\langle e_i, e_j \rangle_{L_2(X; \mu)} = \delta_{ij}, \quad Te_j = \lambda_j e_j.$$

We will call $\lambda_j$ and $e_j$ as eigenvalues and eigenfunctions respectively, and $\{e_j\}_{j \in J}$ clearly forms an ONS. By Mercer's theorem,

$$k(x, y) = \sum_{j \in J} \lambda_j e_j(x) e_j(y), \tag{5}$$

and all functions in $\mathcal{H}$ can be represented by $\sum_{j \in J} a_j e_j$ where $\{a_j / \sqrt{\lambda_j}\} \in \ell^2(J)$. The inner product in $\mathcal{H}$ is equivalent to $\left\langle \sum_{j \in J} a_j e_j, \sum_{j \in J} b_j e_j \right\rangle_{\mathcal{H}} = \sum_{j \in J} a_j b_j / \lambda_j$. Therefore it is easy to see that

$$\varphi_j := \sqrt{\lambda_j} e_j, \qquad j \in J$$

is an orthonormal basis of $\mathcal{H}$, with Moreover, for all $f \in \mathcal{H}$ with $f = \sum_{j \in J} a_j e_j$, we have $\langle f, e_j \rangle_{\mathcal{H}} = a_j / \lambda_j$, $\langle f, \varphi_j \rangle_{\mathcal{H}} = a_j / \sqrt{\lambda_j}$, and

$$f = \sum_j \langle f, \varphi_j \rangle_{\mathcal{H}} \varphi_j = \sum_j \sqrt{\lambda_j} \langle f, e_j \rangle_{\mathcal{H}} \varphi_j = \sum_j \lambda_j \langle f, e_j \rangle_{\mathcal{H}} e_j.$$

Most kernels used in machine learning are infinite dimensional, i.e., $J = \mathbb{N}$. For convenience, we define $\Phi_m := (\varphi_1, \ldots, \varphi_m)$ and $\Lambda_m = \text{diag}(\lambda_1, \ldots, \lambda_m)$.

## A.3 General sample complexity and assumptions on the product kernel

In this section, we first consider kernels $k_0$ with **scalar input**, i.e., $X_0 \subseteq \mathbb{R}$. Assume there is a measure $\mu_0$ on $X_0$. This will serve as the basis for the more general product kernels in the form of $k(x, y) = \prod_{j=1}^{d} k_0(x_j, y_j)$ defined over $X_0^d$.

With Assumptions 1 and 2, we now state the formal version of Theorem 3 by first providing the sample complexity for approximating the partial derivatives. In the next subsection, we will examine how three different kernels satisfy/unsatisfy the Assumptions 1 and 2, and what the value of $N_\varepsilon$ is. For each case, we will specify $\mu_0$ on $X_0$, and the measure on $X_0^d$ is trivially $\mu = \mu_0^d$.

**Theorem 4.** *Suppose $\{w^s\}_{s=1}^n$ are drawn iid from $\mu_0$ on $X_0$, where $\mu_0$ is the uniform distribution on $[-v/2, v/2]$ for periodic kernels or periodized Gaussian kernels. Let $Z := (k_0(w^1, \cdot), k_0(w^2, \cdot), \dots, k_0(w^n, \cdot))$, and $g_1 = \frac{1}{l} \sum_{a=1}^{l} \gamma_a g_1^a \colon X_0^d \to \mathbb{R}$, where $\|\gamma\|_\infty \leq c_1$ and*

$$g_1^a(y) = \partial^{0,1} k(x^a, y) = h_1^a(y_1) \prod_{j=2}^{d} k_0(x_j^a, y_j) \quad \text{with} \quad h_1^a(\cdot) := \partial^{0,1} k_0(x_1^a, \cdot).$$

*Given $\varepsilon \in (0, 1]$, let $\Phi_m = (\varphi_1, \dots \varphi_m)$ where $m = N_\varepsilon$. Then with probability $1 - \delta$, the following holds when the sample size $n = \max(N_\varepsilon, \frac{5}{3\varepsilon^2} N_\varepsilon Q_\varepsilon^2 \log \frac{2N_\varepsilon}{\delta})$:*

$$\|g_1\|_{\mathcal{H}}^2 \leq \frac{1}{l^2} \gamma^\top K_1 \gamma + 3c_1 \left(1 + 2\sqrt{N_\varepsilon} M_\varepsilon\right) \varepsilon, \tag{6}$$

$$\text{where} \quad (K_1)_{a,b} = (h_1^a)^\top Z (Z^\top Z)^{-1} Z^\top h_1^b \prod_{j=2}^{d} k_0(x_j^a, x_j^b).$$

Then we obtain the formal statement of sample complexity, as stated in the following corollary, by combining all the coordinates from Theorem 4.

**Corollary 1.** *Suppose all coordinates share the same set of samples $\{w^s\}_{s=1}^n$. Applying the results in (6) for coordinates from 1 to $d$ and using the union bound, we have that with sample size $n = \max(N_\varepsilon, \frac{5}{3\varepsilon^2} N_\varepsilon Q_\varepsilon^2 \log \frac{2N_\varepsilon}{\delta})$, the following holds with probability $1 - d\delta$,*

$$\lambda_{\max}(G^\top G) \leq \lambda_{\max}(\tilde{P}_G) + 3c_1 \left(1 + 2\sqrt{N_\varepsilon} M_\varepsilon\right) \varepsilon. \tag{7}$$

*Equivalently, if $N_\varepsilon$, $M_\varepsilon$ and $Q_\varepsilon$ are constants or poly-log terms of $\varepsilon$ which we treat as constant, then to ensure $\lambda_{\max}(G^\top G) \leq \lambda_{\max}(\tilde{P}_G) + \varepsilon$ with probability $1 - \delta$, the sample size needs to be*

$$n = \frac{15}{\varepsilon^2} c_1^2 \left(1 + 2\sqrt{N_\varepsilon} M_\varepsilon\right)^2 N_\varepsilon Q_\varepsilon^2 \log \frac{2dN_\varepsilon}{\delta}.$$

**Remark 1.** *The first term on the right-hand side of (7) is explicitly upper bounded by $L^2$ in our training objective. In the case of Theorem 1, the values of $Q_\varepsilon$, $N_\varepsilon$, and $M_\varepsilon$ lead to a $\tilde{O}(\frac{1}{\varepsilon^2})$ sample complexity. If we further zoom into the dependence on the period $v$, then note that $N_\varepsilon$ is almost a universal constant while $M_\varepsilon = \frac{\sqrt{2}\pi}{v}(N_\varepsilon - 1)$. So overall, $n$ depends on $v$ by $\frac{1}{v^2}$. This is not surprising because smaller period means higher frequency, hence more samples are needed.*

**Remark 2.** *Corollary 1 postulates that all coordinates share the same set of samples $\{w^s\}_{s=1}^n$. When coordinates differ in their domains, we can draw different sets of samples for them. The sample complexity hence grows by $d$ times as we only use a weak union bound. More refined analysis could save us a factor of $d$ as these sets of samples are independent of each other.*

*Proof of Theorem 4.* Let $\varepsilon' := (1 + 2\sqrt{m} M_\varepsilon)\varepsilon$. Since $\langle g_1^a, g_1^b \rangle_{\mathcal{H}} = \langle h_1^a, h_1^b \rangle_{\mathcal{H}_0} \prod_{j=2}^{d} k_0(x_j^a, x_j^b)$ and $|k_0(x_j^a, x_j^b)| \leq 1$, it suffices to show that for all $a, b \in [l]$,

$$\left| \langle h_1^a, h_1^b \rangle_{\mathcal{H}_0} - (h_1^a)^\top Z (Z^\top Z)^{-1} Z^\top h_1^b \right| \leq 3\varepsilon'.$$

Towards this end, it is sufficient to show that for any $h(\cdot) = \vartheta_x \partial^{0,1} k_0(x, \cdot) + \vartheta_y \partial^{0,1} k_0(y, \cdot)$ where $x, y \in X_0$ and $|\vartheta_x| + |\vartheta_y| \leq 1$, we have

$$\left| h^\top Z (Z^\top Z)^{-1} Z^\top h - \|h\|_{\mathcal{H}_0}^2 \right| \leq \varepsilon'. \tag{8}$$

This is because, if so, then

$$
\begin{aligned}
&\left| \langle h_1^a, h_1^b \rangle_{\mathcal{H}_0} - (h_1^a)^\top Z(Z^\top Z)^{-1} Z^\top h_1^b \right| \\
&= \left| \frac{1}{2} \left( \left\| h_1^a + h_1^b \right\|_{\mathcal{H}_0}^2 - \left\| h_1^a \right\|_{\mathcal{H}_0}^2 - \left\| h_1^b \right\|_{\mathcal{H}_0}^2 \right) - \frac{1}{2} \left[ (h_1^a + h_1^b)^\top Z(Z^\top Z)^{-1} Z^\top (h_1^a + h_1^b) \right. \right. \\
&\qquad \left. \left. - (h_1^a)^\top Z(Z^\top Z)^{-1} Z^\top h_1^a - (h_1^b)^\top Z(Z^\top Z)^{-1} Z^\top h_1^b \right] \right| \\
&\leq \frac{1}{2}(4\varepsilon' + \varepsilon' + \varepsilon') = 3\varepsilon'.
\end{aligned}
$$

The rest of the proof is devoted to (8). Since $n \geq m$, the SVD of $\Lambda_m^{-1/2} \Phi_m^\top Z$ can be written as $U\Sigma V^\top$, where $UU^\top = U^\top U = V^\top V = I_m$ ($m$-by-$m$ identity matrix), and $\Sigma = \mathrm{diag}(\sigma_1, \ldots, \sigma_m)$. Define

$$
\boldsymbol{\alpha} = n^{-1/2} V U^\top \Lambda_m^{-1/2} \Phi_m^\top h.
$$

Consider the optimization problem $o(\boldsymbol{\alpha}) := \frac{1}{2} \| Z\boldsymbol{\alpha} - h \|_{\mathcal{H}_0}^2$. It is easy to see that its minimal objective value is $o^* := \frac{1}{2} \| h \|_{\mathcal{H}_0}^2 - \frac{1}{2} h^\top Z(Z^\top Z)^{-1} Z^\top h$. So

$$
0 \leq 2o^* = \| h \|_{\mathcal{H}_0}^2 - h^\top Z(Z^\top Z)^{-1} Z^\top h \leq 2o(\boldsymbol{\alpha}).
$$

Therefore to prove (8), it suffices to bound $o(\boldsymbol{\alpha}) = \| Z\boldsymbol{\alpha} - h \|_{\mathcal{H}_0}$. Since $\sqrt{n} \Phi_m \Lambda^{1/2} U V^\top \boldsymbol{\alpha} = \Phi_m \Phi_m^\top h$, we can decompose $\| Z\boldsymbol{\alpha} - h \|_{\mathcal{H}_0}$ by

$$
\| Z\boldsymbol{\alpha} - h \|_{\mathcal{H}_0} \leq \left\| (Z - \Phi_m \Phi_m^\top Z)\boldsymbol{\alpha} \right\|_{\mathcal{H}_0} + \left\| (\Phi_m \Phi_m^\top Z - \sqrt{n} \Phi_m \Lambda_m^{1/2} U V^\top)\boldsymbol{\alpha} \right\|_{\mathcal{H}_0} \tag{9}
$$
$$
+ \left\| \Phi_m \Phi_m^\top h - h \right\|_{\mathcal{H}_0}.
$$

The last term $\left\| \Phi_m \Phi_m^\top h - h \right\|_{\mathcal{H}_0}$ is clearly below $\varepsilon$ because by Assumption 1 and $m = N_\varepsilon$

$$
\begin{aligned}
&\left\| \Phi_m \Phi_m^\top h - h \right\|_{\mathcal{H}_0} \\
&\leq |\vartheta_x| \left\| \Phi_m \Phi_m^\top \partial^{0,1} k_0(x, \cdot) - \partial^{0,1} k_0(x, \cdot) \right\|_{\mathcal{H}_0} + |\vartheta_y| \left\| \Phi_m \Phi_m^\top \partial^{0,1} k_0(y, \cdot) - \partial^{0,1} k_0(y, \cdot) \right\|_{\mathcal{H}_0} \\
&\leq (|\vartheta_x| + |\vartheta_y|) \varepsilon \leq \varepsilon.
\end{aligned}
$$

We will next bound the first two terms on the right-hand side of (9).

(i) By Assumption 1, $\left\| k_0(w^s, \cdot) - \Phi_m \Phi_m^\top k_0(w^s, \cdot) \right\|_{\mathcal{H}_0} \leq \varepsilon$, hence $\left\| (Z - \Phi_m \Phi_m^\top Z)\boldsymbol{\alpha} \right\|_{\mathcal{H}_0} \leq \varepsilon \sqrt{n} \| \boldsymbol{\alpha} \|_2$. To bound $\| \boldsymbol{\alpha} \|_2$, note all singular values of $VU^\top$ are 1, and so Assumption 2 implies that for all $i \in [m]$,

$$
\left| \lambda_j^{-1/2} \langle \varphi_j, h \rangle_{\mathcal{H}_0} \right| = \left| \langle e_j, h \rangle_{\mathcal{H}_0} \right| = \left| \langle e_j, \vartheta_x \partial^{0,1} k_0(x, \cdot) + \vartheta_y \partial^{0,1} k_0(y, \cdot) \rangle_{\mathcal{H}_0} \right| \tag{10}
$$
$$
\leq \sup_{x \in X} \left| \langle e_j, \partial^{0,1} k(x, \cdot) \rangle_{\mathcal{H}_0} \right| \leq M_\varepsilon.
$$

As a result,

$$
\left\| (Z - \Phi_m \Phi_m^\top Z)\boldsymbol{\alpha}_j \right\|_{\mathcal{H}_0} \leq \varepsilon n^{1/2} \cdot n^{-1/2} \left\| \Lambda_m^{-1/2} \Phi_m^\top h \right\| \leq \varepsilon \sqrt{m} M_\varepsilon.
$$

(ii) We first consider the concentration of the matrix $R := \frac{1}{n} \Lambda_m^{-1/2} \Phi_m^\top Z Z^\top \Phi_m \Lambda_m^{-1/2} \in \mathbb{R}^{m \times m}$. Clearly,

$$
\mathop{\mathbb{E}}_{\{w_s\}} [R_{ij}] = \mathop{\mathbb{E}}_{\{w_s\}} \left[ \frac{1}{n} \sum_{s=1}^n e_i(w_s) e_j(w_s) \right] = \int e_i(x) e_j(x) \, \mathrm{d}\mu(x) = \delta_{ij}.
$$

By matrix Bernstein theorem (Tropp, 2015, Theorem 1.6.2), we have $\Pr\left( \| R - I_m \|_{sp} \leq \varepsilon \right) \geq 1 - \delta$ when $n \geq O(.)$. This is because $\| (e_1(x), \ldots, e_m(x)) \|^2 \leq m Q_\varepsilon^2$, $\left\| \mathbb{E}_{\{w_s\}}[RR^\top] \right\|_{sp} \leq m Q_\varepsilon^2 / n$, and

$$
\Pr\left( \| R - I_m \|_{sp} \leq \varepsilon \right) \geq 1 - 2m \exp\left( \frac{-\varepsilon^2}{\frac{m Q_\varepsilon^2}{n}\left(1 + \frac{2}{3}\varepsilon\right)} \right) \geq 1 - 2m \exp\left( \frac{-\varepsilon^2}{\frac{5 m Q_\varepsilon^2}{3n}} \right) \geq 1 - \delta,
$$

where the last step is by the definition of $n$. Since $R = \frac{1}{n} U \Sigma^2 U^\top$, this means with probability $1 - \delta$, $\left\| \frac{1}{n} U \Sigma^2 U^\top - I_m \right\|_{sp} \leq \varepsilon$. So for all $i \in [m]$,

$$\left| \frac{1}{n} \sigma_i^2 - 1 \right| \leq \varepsilon \qquad \text{which implies} \qquad \left| \frac{1}{\sqrt{n}} \sigma_i - 1 \right| < \varepsilon \left| \frac{1}{\sqrt{n}} \sigma_i + 1 \right|^{-1} \leq \varepsilon. \tag{11}$$

Moreover, $\lambda_1 \leq 1$ since $k_0(x, x) = 1$. It then follows that

$$
\begin{aligned}
&\left\| (\Phi_m \Phi_m^\top Z - \sqrt{n} \Phi_m \Lambda_m^{1/2} U V^\top) \boldsymbol{\alpha} \right\|_{\mathcal{H}_0} \\
&= \left\| \Phi_m \Lambda_m^{1/2} U \Sigma V^\top \frac{1}{\sqrt{n}} V U^\top \Lambda_m^{-1/2} \Phi_m^\top h - \sqrt{n} \Phi_m \Lambda_m^{1/2} U V^\top \frac{1}{\sqrt{n}} V U^\top \Lambda_m^{-1/2} \Phi_m^\top h \right\|_{\mathcal{H}_0} \\
&= \left\| \Lambda_m^{1/2} U \left( \frac{1}{\sqrt{n}} \Sigma - I_m \right) U^\top \Lambda_m^{-1/2} \Phi_m^\top h \right\|_2 \qquad \text{(because } \Phi_m^\top \Phi_m = I_m) \\
&\leq \sqrt{\lambda_1} \max_{i \in [m]} \left| \frac{1}{\sqrt{n}} \sigma_i - 1 \right| \left\| \Lambda_m^{-1/2} \Phi_m^\top h \right\|_2 \\
&\leq \varepsilon \sqrt{m} M_\varepsilon \qquad \text{(by (11), (10), and } \lambda_1 \leq 1).
\end{aligned}
$$

Combining (i) and (ii), we arrive at the desired bound in (6). ∎

*Proof of Corollary 1.* Since $\tilde{P}_G$ approximates $G^\top G$ only on the diagonal, $\tilde{P}_G - G^\top G$ is a diagonal matrix which we denote as $\mathrm{diag}(\delta_1, \ldots, \delta_d)$. Let $\boldsymbol{u} \in \mathbb{R}^d$ be the leading eigenvector of $\tilde{P}_G$. Then

$$\lambda_{\max}(\tilde{P}_G) - \lambda_{\max}(G^\top G) \leq \boldsymbol{u}^\top \tilde{P}_G \boldsymbol{u} - \boldsymbol{u}^\top G^\top G \boldsymbol{u} = \boldsymbol{u}^\top (\tilde{P}_G - G^\top G) \boldsymbol{u} = \sum_j \delta_j \boldsymbol{u}_j^2$$

$$\text{(by (6))} \quad \leq 3 c_1 \left( 1 + 2 \sqrt{N_\varepsilon} M_\varepsilon \right) \varepsilon.$$

The proof is completed by applying the union bound and rewriting the results. ∎

### A.4 CASE 1: CHECKING ASSUMPTIONS 1 AND 2 ON PERIODIC KERNELS

Periodic kernels on $X_0 := \mathbb{R}$ are translation invariant, and can be written as $k_0(x, y) = \kappa(x - y)$ where $\kappa : \mathbb{R} \to \mathbb{R}$ is a) periodic with period $v$; b) even, with $\kappa(-t) = \kappa(t)$; and c) normalized with $\kappa(0) = 1$. A general treatment was given by Williamson et al. (2001), and an example was given by David MacKay in MacKay (1998):

$$k_0(x, y) = \exp \left( -\frac{1}{2\sigma^2} \sin \left( \frac{\pi}{v} (x - y) \right)^2 \right). \tag{12}$$

We define $\mu_0$ to be a uniform distribution on $[-\frac{v}{2}, \frac{v}{2}]$, and let $\omega_0 = 2\pi/v$.

Since $\kappa$ is symmetric, we can simplify the Fourier transform of $\kappa(t) \delta_v(t)$, where $\delta_v(t) = 1$ if $t \in [-v/2, v/2]$, and 0 otherwise:

$$F(\omega) = \frac{1}{\sqrt{2\pi}} \int_{-v/2}^{v/2} \kappa(t) \cos(\omega t) \, \mathrm{d}t.$$

It is now easy to observe that thanks to periodicity and symmetry of $\kappa$, for all $j \in \mathbb{Z}$,

$$\frac{1}{v} \int_{-v/2}^{v/2} k_0(x, y) \cos(j\omega_0 y) \, \mathrm{d}y = \frac{1}{v} \int_{-v/2}^{v/2} \kappa(x - y) \cos(j\omega_0 y) \, \mathrm{d}y$$

$$= \frac{1}{v} \int_{x-v/2}^{x+v/2} \kappa(z) \cos(j\omega_0(x - z)) \, \mathrm{d}z \quad (\text{note } \cos(j\omega_0(x - z)) \text{ also has period } v)$$

$$= \frac{1}{v} \int_{-v/2}^{v/2} \kappa(z)[\cos(j\omega_0 x) \cos(j\omega_0 z) + \sin(j\omega_0 x) \sin(j\omega_0 z)] \, \mathrm{d}z \quad (\text{by periodicity})$$

$$= \frac{1}{v} \cos(j\omega_0 x) \int_{-v/2}^{v/2} \kappa(z) \cos(j\omega_0 z) \, \mathrm{d}z \quad (\text{by symmetry of } \kappa)$$

$$= \frac{\sqrt{2\pi}}{v} F(j\omega_0) \cos(j\omega_0 x).$$

And similarly,

$$\frac{1}{v} \int_{-v/2}^{v/2} k_0(x, y) \sin(j\omega_0 y) \, \mathrm{d}y = \frac{\sqrt{2\pi}}{v} F(j\omega_0) \sin(j\omega_0 x).$$

Therefore the eigenfunctions of the integral operator $T_k$ are

$$e_0(x) = 1, \quad e_j(x) := \sqrt{2} \cos(j\omega_0 x), \quad e_{-j}(x) := \sqrt{2} \sin(j\omega_0 x) \quad (j \geq 1)$$

and the eigenvalues are $\lambda_j = \frac{\sqrt{2\pi}}{v} F(j\omega_0)$ for all $j \in \mathbb{Z}$ with $\lambda_{-j} = \lambda_j$. An important property our proof will rely on is that

$$e'_j(x) = -j\omega_0 e_{-j}(x), \quad \text{for all } j \in \mathbb{Z}.$$

Applying Mercer's theorem in (5) and noting $\kappa(0) = 1$, we derive $\sum_{j \in \mathbb{Z}} \lambda_j = 1$.

**Checking the Assumptions 1 and 2.** The following theorem summarizes the assumptions and conclusions regarding the satisfaction of Assumptions 1 and 2. Again we focus on the case of $X \subseteq \mathbb{R}$.

**Theorem 1.** *Suppose the periodic kernel with period $v$ has eigenvalues $\lambda_j$ that satisfies*

$$\lambda_j(1 + j)^2 \max(1, j^2)(1 + \delta(j \geq 1)) \leq c_6 \cdot c_4^{-j}, \quad \text{for all } j \geq 0, \tag{13}$$

*where $c_4 > 1$ and $c_6 > 0$ are universal constants. Then Assumption 1 holds with*

$$N_\varepsilon = 1 + 2 \lfloor n_\varepsilon \rfloor, \quad \text{where} \quad n_\varepsilon := \log_{c_4}\left(\frac{2.1 c_6}{\varepsilon^2} \max\left(1, \frac{v^2}{4\pi^2}\right)\right). \tag{14}$$

*In addition, Assumption 2 holds with $Q_\varepsilon = \sqrt{2}$ and $M_\varepsilon = \frac{2\sqrt{2\pi}}{v} \lfloor n_\varepsilon \rfloor = \frac{\sqrt{2\pi}}{v}(N_\varepsilon - 1)$.*

For example, if we set $v = \pi$ and $\sigma^2 = 1/2$ in the kernel in (12), elementary calculation shows that the condition (13) is satisfied with $c_4 = 2$ and $c_6 = 1.6$.

*Proof of Theorem 1.* First we show that $h(x) := \partial^{0,1} k_0(x_0, x)$ is in $\mathcal{H}_0$ for all $x_0 \in X_0$. Since $k_0(x_0, x) = \sum_{j \in \mathbb{Z}} \lambda_j e_j(x_0) e_j(x)$, we derive

$$h(x) = \sum_{j \in \mathbb{Z}} \lambda_j e_j(x_0) \partial^1 e_j(x) = \sum_{j \in \mathbb{Z}} \lambda_j e_j(x_0)(-j\omega_0 e_{-j}(x)) = \omega_0 \sum_{j \in \mathbb{Z}} \lambda_j j e_{-j}(x_0) e_j(x). \tag{15}$$

$h(x)$ is in $\mathcal{H}$ if the sequence $\lambda_j j e_{-j}(x_0)/\sqrt{\lambda_j}$ is square summable. This can be easily seen by (13):

$$\omega_0^{-2} \|h\|_{\mathcal{H}_0}^2 = \sum_j \lambda_j j^2 e_{-j}^2(x_0) = \sum_{j \in \mathbb{Z}} \lambda_j j^2 e_{-j}^2(x_0)$$

$$= \sum_{j \in \mathbb{Z}} \lambda_j j^2 e_{-j}^2(x_0) = \lambda_0 + 2 \sum_{j \geq 1} j^2 \lambda_j \leq \frac{2 c_4 c_5}{c_4 - 1}.$$

Finally to derive $N_\varepsilon$, we reuse the orthonormal decomposition of $h(x)$ in (15). For a given set of $j$ values $A$ where $A \subseteq \mathbb{Z}$, we denote as $\Phi_A$ the "matrix" whose columns enumerate the $\varphi_j$ over $j \in A$. Let us choose

$$A := \left\{ j : \lambda_j \max(1, j^2)(1 + j^2)(1 + \delta(j \geq 1)) \geq \min(1, w_0^{-2}) \frac{\varepsilon^2}{2.1} \right\}.$$

If $j \in A$, then $-j \in A$. Letting $\mathbb{N}_0 = \{0, 1, 2, \ldots\}$, we note $\sum_{j \in \mathbb{N}_0} \frac{1}{1+j^2} \leq 2.1$. So

$$\begin{aligned}
\left\| h - \Phi_A \Phi_A^\top h \right\|_{\mathcal{H}_0}^2 &= w_0^2 \sum_{j \in \mathbb{Z} \setminus A} \lambda_j j^2 e_{-j}^2(x_0) \\
&= w_0^2 \sum_{j \in \mathbb{N}_0 \setminus A} \lambda_j j^2 \left[ (e_j^2(x) + e_{-j}^2(x)) \delta(j \geq 1) + \delta(j = 0) \right] \\
&= w_0^2 \sum_{j \in \mathbb{N}_0 \setminus A} \lambda_j j^2 (1 + \delta(j \geq 1)) \\
&= w_0^2 \sum_{j \in \mathbb{N}_0 \setminus A} \left\{ \lambda_j j^2 (1 + j^2)(1 + \delta(j \geq 1)) \frac{1}{1 + j^2} \right\} \\
&\leq \frac{\varepsilon^2}{2.1} \sum_{j \in \mathbb{N}_0} \frac{1}{1 + j^2} = \frac{\varepsilon^2}{2.1} \sum_{j \in \mathbb{N}_0} \frac{1}{1 + j^2} \leq \varepsilon^2.
\end{aligned}$$

Similarly, we can bound $\left\| k_0(x_0, \cdot) - \Phi_A \Phi_A^\top k_0(x_0, \cdot) \right\|_{\mathcal{H}_0}$ by

$$\begin{aligned}
& \left\| k_0(x_0, \cdot) - \Phi_A \Phi_A^\top k_0(x_0, \cdot) \right\|_{\mathcal{H}_0}^2 \\
&= \sum_{j \in \mathbb{Z} \setminus A} \lambda_j e_j^2(x_0) \leq \sum_{j \in \mathbb{Z} \setminus A} \lambda_j \max(1, j^2) e_j^2(x_0) \\
&= \sum_{j \in \mathbb{N}_0 \setminus A} \lambda_{\boldsymbol{\alpha}} \max(1, j^2) [(e_j^2(x) + e_{-j}^2(x)) \delta(j \geq 1) + \delta(j = 0)] \\
&= \sum_{j \in \mathbb{N}_0 \setminus A} \left\{ \lambda_j \max(1, j^2)(1 + j^2)(1 + \delta(j \geq 1)) \frac{1}{1 + j^2} \right\} \\
&\leq \frac{1}{2.1} \varepsilon^2 \sum_{j \in \mathbb{N}_0} \frac{1}{1 + j^2} \leq \varepsilon^2.
\end{aligned}$$

To upper bound the cardinality of $A$, we consider the conditions for $j \notin A$. Thanks to the conditions in (13), we know that any $j$ satisfying the following relationship cannot be in $A$:

$$c_6 \cdot c_4^{-|j|} < \min(1, w_0^{-2}) \frac{\varepsilon^2}{2.1} \qquad \Leftrightarrow \qquad c_4^{-|j|} < \frac{1}{2.1 \cdot c_6} \min\left(1, \frac{4\pi^2}{v^2}\right) \varepsilon^2.$$

So $A \subseteq \{j : |j| \leq n_\varepsilon\}$, which yields the conclusion (14). Finally $Q_\varepsilon \leq \sqrt{2}$, and to bound $M_\varepsilon$, we simply reuse (15). For any $j$ with $|j| \leq n_\varepsilon$,

$$\left| \langle h, e_j \rangle_{\mathcal{H}} \right| \leq \omega_0 |j e_{-j}(x_0)| \leq \frac{2\pi}{v} \sqrt{2} \lfloor n_\varepsilon \rfloor = \frac{\sqrt{2}\pi}{v} (N_\varepsilon - 1). \qquad \blacksquare$$

## A.5 CASE 2: CHECKING ASSUMPTIONS 1 AND 2 ON GAUSSIAN KERNELS

Gaussian kernels $k(x, y) = \exp(-\|x - y\|^2 / (2\sigma^2))$ are obviously product kernels with $k_0(x_1, y_1) = \kappa(x_1 - y_1) = \exp(-(x_1 - y_1)^2 / (2\sigma^2))$. It is also translation invariant. The spectrum of Gaussian kernel $k_0$ on $\mathbb{R}$ is known; see, e.g., Chapter 4.3.1 of Rasmussen & Williams (2006) and Section 4 of Zhu et al. (1998). Let $\mu$ be a Gaussian distribution $\mathcal{N}(0, \sigma^2)$. Setting $\varepsilon^2 = \alpha^2 = (2\sigma^2)^{-1}$

in Eq 12 and 13 of E Fasshauer (2011), the eigenvalue and eigenfunctions are (for $j \geq 0$):

$$\lambda_j = c_0^{-j-1/2}, \quad \text{where} \quad c_0 = \frac{1}{2}(3 + \sqrt{5})$$

$$e_j(x) = \frac{5^{1/8}}{2^{j/2}} \exp\left(-\frac{\sqrt{5}-1}{4}\frac{x^2}{\sigma^2}\right) \frac{1}{\sqrt{j!}} H_j\left(\sqrt[4]{1.25}\,\frac{x}{\sigma}\right),$$

where $H_j$ is the Hermite polynomial of order $j$.

Although the eigenvalues decay exponentially fast, the eigenfunctions are not uniformly bounded in the $L_\infty$ sense. Although the latter can be patched if we restrict $x$ to a bounded set, the above closed-form of eigen-pairs will no longer hold, and the analysis will become rather challenging.

To resolve this issue, we resort to the period-ization technique proposed by Williamson et al. (2001). Consider $\kappa(x) = \exp(-x^2/(2\sigma^2))$ when $x \in [-v/2, v/2]$, and then extend $\kappa$ to $\mathbb{R}$ as a periodic function with period $v$. Again let $\mu$ be the uniform distribution on $[-v/2, v/2]$. As can be seen from the discriminant function $f = \frac{1}{l}\sum_{i=1}^{l} \gamma_i k(x^i, \cdot)$, as along as our training and test data both lie in $[-v/4, v/4]$, the modification of $\kappa$ outside $[-v/2, v/2]$ does not effectively make any difference. Although the term $\partial^{0,1} k_0(x_1^a, w_1^1)$ in (13) may possibly evaluate $\kappa$ outside $[-v/2, v/2]$, it is only used for testing the gradient norm bound of $\kappa$.

With this periodized Gaussian kernel, it is easy to see that $Q_\varepsilon = \sqrt{2}$. If we standardize by $\sigma = 1$ and set $v = 5\pi$ as an example, it is not hard to see that (13) holds with $c_4 = 1.25$ and $c_6 = 50$. The expressions of $N_\varepsilon$ and $M_\varepsilon$ then follow from Theorem 1 directly.

### A.6    CASE 3: CHECKING ASSUMPTIONS 1 AND 2 ON NON-PRODUCT KERNELS

The above analysis has been restricted to product kernels. But in practice, there are many useful kernels that are not decomposable. A prominent example is the inverse kernel: $k(x, y) = (2 - x^\top y)^{-1}$. In general, it is extremely challenging to analyze eigenfunctions, which are commonly *not* bounded (Lafferty & Lebanon, 2005; Zhou, 2002), i.e., $\sup_{i \to \infty} \sup_x |e_i(x)| = \infty$. The opposite was (incorrectly) claimed in Theorem 4 of Williamson et al. (2001) by citing an incorrect result in König (1986, p. 145), which was later corrected by Zhou (2002) and Steve Smale. Indeed, uniform boundedness is not known even for Gaussian kernels with uniform distribution on $[0, 1]^d$ Lin et al. (2017), and (Minh et al., 2006, Theorem 5) showed the unboundedness for Gaussian kernels with uniform distribution on the unit sphere when $d \geq 3$.

Here we only present the limited results that we have obtained on the eigenvalues of the integral operator of inverse kernels with a uniform distribution on the unit ball. The analysis of eigenfunctions is left for future work. Specifically, in order to drive the eigenvalue $\lambda_i$ below $\varepsilon$, $i$ must be at least $d^{\lceil \log_2 \frac{1}{\varepsilon} \rceil + 1}$. This is a quasi-quadratic bound if we view $d$ and $1/\varepsilon$ as two large variables.

It is quite straightforward to give an explicit characterization of the functions in $\mathcal{H}$. The Taylor expansion of $z^{-1}$ at $z = 2$ is $\frac{1}{2}\sum_{i=0}^{\infty}(-\frac{1}{2})^i x^i$. Using the standard multi-index notation with $\boldsymbol{\alpha} = (\alpha_1, \ldots, \alpha_d) \in (\mathbb{N} \cup \{0\})^d$, $|\boldsymbol{\alpha}| = \sum_{i=1}^{d} \alpha_i$, and $\boldsymbol{x}^{\boldsymbol{\alpha}} = x_1^{\alpha_1}\ldots x_d^{\alpha_d}$, we derive

$$k(\boldsymbol{x}, \boldsymbol{y}) = \frac{1}{2 - \boldsymbol{x}^\top \boldsymbol{y}} = \frac{1}{2}\sum_{k=0}^{\infty}\left(-\frac{1}{2}\right)^k (-\boldsymbol{x}^\top \boldsymbol{y})^k = \sum_{k=0}^{\infty} 2^{-k-1} \sum_{\boldsymbol{\alpha}:|\boldsymbol{\alpha}|=k} C_{\boldsymbol{\alpha}}^k \boldsymbol{x}^{\boldsymbol{\alpha}} \boldsymbol{y}^{\boldsymbol{\alpha}}$$

$$= \sum_{\boldsymbol{\alpha}} 2^{-|\boldsymbol{\alpha}|-1} C_{\boldsymbol{\alpha}}^{|\boldsymbol{\alpha}|} \boldsymbol{x}^{\boldsymbol{\alpha}} \boldsymbol{y}^{\boldsymbol{\alpha}},$$

where $C_{\boldsymbol{\alpha}}^k = \frac{k!}{\prod_{i=1}^{d} \alpha_i!}$. So we can read off the feature mapping for $\boldsymbol{x}$ as

$$\varphi(\boldsymbol{x}) = \{w_{\boldsymbol{\alpha}} \boldsymbol{x}^{\boldsymbol{\alpha}} : \boldsymbol{\alpha}\}, \quad \text{where} \quad w_{\boldsymbol{\alpha}} = 2^{-\frac{1}{2}(|\boldsymbol{\alpha}|+1)} C_{\boldsymbol{\alpha}}^{|\boldsymbol{\alpha}|},$$

and the functions in $\mathcal{H}$ are

$$\mathcal{H} = \left\{ f = \sum_{\boldsymbol{\alpha}} \vartheta_{\boldsymbol{\alpha}} w_{\boldsymbol{\alpha}} \boldsymbol{x}^{\boldsymbol{\alpha}} : \|\vartheta\|_{\ell_2} < \infty \right\}. \tag{16}$$

Note this is just an intuitive "derivation" while a rigorous proof for (16) can be constructed in analogy to that of Theorem 1 in Minh (2010).

### A.7 Background of eigenvalues of a kernel

We now use (16) to find the eigenvalues of inverse kernel.

Now specializing to our inverse kernel case, let us endow a uniform distribution over the unit ball $B$: $p(x) = V_d^{-1}$ where $V_d = \pi^{d/2} \Gamma(\frac{d}{2} + 1)^{-1}$ is the volume of $B$, with $\Gamma$ being the Gamma function. Then $\lambda$ is an eigenvalue of the kernel if there exists $f = \sum_{\boldsymbol{\alpha}} \vartheta_{\boldsymbol{\alpha}} w_{\boldsymbol{\alpha}} x^{\boldsymbol{\alpha}}$ such that $\int_{\boldsymbol{y} \in B} k(\boldsymbol{x}, \boldsymbol{y}) p(\boldsymbol{y}) f(\boldsymbol{y}) \, \mathrm{d}\boldsymbol{y} = \lambda f(\boldsymbol{x})$. This translates to

$$V_d^{-1} \int_{\boldsymbol{y} \in B} \sum_{\boldsymbol{\alpha}} w_{\boldsymbol{\alpha}}^2 \boldsymbol{x}^{\boldsymbol{\alpha}} \boldsymbol{y}^{\boldsymbol{\alpha}} \sum_{\boldsymbol{\beta}} \vartheta_{\boldsymbol{\beta}} w_{\boldsymbol{\beta}} \boldsymbol{y}^{\boldsymbol{\beta}} \, \mathrm{d}\boldsymbol{y} = \lambda \sum_{\boldsymbol{\alpha}} \vartheta_{\boldsymbol{\alpha}} w_{\boldsymbol{\alpha}} \boldsymbol{x}^{\boldsymbol{\alpha}}, \qquad \forall \, \boldsymbol{x} \in B.$$

Since $B$ is an open set, that means

$$w_{\boldsymbol{\alpha}} \sum_{\boldsymbol{\beta}} w_{\boldsymbol{\beta}} q_{\boldsymbol{\alpha}+\boldsymbol{\beta}} \vartheta_{\boldsymbol{\beta}} = \lambda \vartheta_{\boldsymbol{\alpha}}, \qquad \forall \, \boldsymbol{\alpha},$$

where

$$q_{\boldsymbol{\alpha}} = V_d^{-1} \int_{\boldsymbol{y} \in B} \boldsymbol{y}^{\boldsymbol{\alpha}} \, \mathrm{d}\boldsymbol{y} = \begin{cases} \dfrac{2 \prod_{i=1}^d \Gamma\left(\frac{1}{2}\alpha_i + \frac{1}{2}\right)}{V_d \cdot (|\boldsymbol{\alpha}| + d) \cdot \Gamma\left(\frac{1}{2}|\boldsymbol{\alpha}| + \frac{d}{2}\right)} & \text{if all } \alpha_i \text{ are even} \\ 0 & \text{otherwise} \end{cases}.$$

In other words, $\lambda$ is the eigenvalue of the infinite dimensional matrix $Q = [w_{\boldsymbol{\alpha}} w_{\boldsymbol{\beta}} q_{\boldsymbol{\alpha}+\boldsymbol{\beta}}]_{\boldsymbol{\alpha}, \boldsymbol{\beta}}$,

### A.8 Bounding the eigenvalues

To bound the eigenvalues of $Q$, we resort to the majorization results in matrix analysis. Since $k$ is a PSD kernel, all its eigenvalues are nonnegative, and suppose they are sorted decreasingly as $\lambda_1 \geq \lambda_2 \geq \dots$. Let the row corresponding to $\boldsymbol{\alpha}$ have $\ell_2$ norm $r_{\boldsymbol{\alpha}}$, and let them be sorted as $r_{[1]} \geq r_{[2]} \geq \dots$. Then by Schneider (1953); Shi & Wang (1965), we have

$$\prod_{i=1}^n \lambda_i \leq \prod_{i=1}^n r_{[i]}, \quad \forall \, n \geq 1.$$

So our strategy is to bound $r_{\boldsymbol{\alpha}}$ first. To start with, we decompose $q_{\boldsymbol{\alpha}+\boldsymbol{\beta}}$ into $q_{\boldsymbol{\alpha}}$ and $q_{\boldsymbol{\beta}}$ via Cauchy-Schwartz:

$$q_{\boldsymbol{\alpha}+\boldsymbol{\beta}}^2 = V_d^{-2} \left( \int_{\boldsymbol{y} \in B} \boldsymbol{y}^{\boldsymbol{\alpha}+\boldsymbol{\beta}} \, \mathrm{d}\boldsymbol{y} \right)^2 \leq V_d^{-2} \int_{\boldsymbol{y} \in B} \boldsymbol{y}^{2\boldsymbol{\alpha}} \, \mathrm{d}\boldsymbol{y} \cdot \int_{\boldsymbol{y} \in B} \boldsymbol{y}^{2\boldsymbol{\beta}} \, \mathrm{d}\boldsymbol{y} = q_{2\boldsymbol{\alpha}} q_{2\boldsymbol{\beta}}.$$

To simplify notation, we consider without loss of generality that $d$ is an even number, and denote the integer $b := d/2$. Now $V_d = \pi^b / b!$. Noting that there are $\begin{pmatrix} k + d - 1 \\ k \end{pmatrix}$ values of $\boldsymbol{\beta}$ such that $|\boldsymbol{\beta}| = k$, we can proceed by (fix below by changing $\begin{pmatrix} k + d \\ k \end{pmatrix}$ into $\begin{pmatrix} k + d - 1 \\ k \end{pmatrix}$, or no need because the former upper bounds the latter)

$$r_{\boldsymbol{\alpha}}^2 = w_{\boldsymbol{\alpha}}^2 \sum_{\boldsymbol{\beta}} w_{\boldsymbol{\beta}}^2 q_{\boldsymbol{\alpha}+\boldsymbol{\beta}}^2 \leq w_{\boldsymbol{\alpha}}^2 q_{2\boldsymbol{\alpha}} \sum_{\boldsymbol{\beta}} w_{\boldsymbol{\beta}}^2 q_{2\boldsymbol{\beta}} = w_{\boldsymbol{\alpha}}^2 q_{2\boldsymbol{\alpha}} \sum_{k=0}^{\infty} 2^{-k-1} \sum_{\boldsymbol{\beta}: |\boldsymbol{\beta}|=k} C_{\boldsymbol{\beta}}^k q_{2\boldsymbol{\beta}}$$

$$\leq w_{\boldsymbol{\alpha}}^2 q_{2\boldsymbol{\alpha}} \sum_{k=0}^{\infty} 2^{-k-1} \begin{pmatrix} k + d \\ d \end{pmatrix} \max_{|\boldsymbol{\beta}|=k} C_{\boldsymbol{\beta}}^k q_{2\boldsymbol{\beta}}$$

$$= w_{\boldsymbol{\alpha}}^2 q_{2\boldsymbol{\alpha}} \sum_{k=0}^{\infty} 2^{-k-1} \begin{pmatrix} k + d \\ d \end{pmatrix} \max_{|\boldsymbol{\beta}|=k} \frac{k!}{\prod_{i=1}^d \beta_i!} \cdot \frac{2 \prod_{i=1}^d \Gamma(\beta_i + \frac{1}{2})}{V_d \cdot (2k + d) \cdot \Gamma(k + \frac{d}{2})}$$

$$= w_{\boldsymbol{\alpha}}^2 q_{2\boldsymbol{\alpha}} V_d^{-1} \sum_{k=0}^{\infty} 2^{-k} \begin{pmatrix} k + d \\ d \end{pmatrix} \frac{k!}{(2k + d) \Gamma(k + \frac{d}{2})} \cdot \max_{|\boldsymbol{\beta}|=k} \prod_{i=1}^d \frac{\Gamma(\beta_i + \frac{1}{2})}{\beta_i!}$$

$$< w_{\boldsymbol{\alpha}}^2 q_{2\boldsymbol{\alpha}} \cdot \frac{b!}{\pi^b d!} \cdot \sum_{k=0}^{\infty} 2^{-k-1} \frac{(k + d)!}{(k + b)!} \qquad (\text{since } \Gamma(\beta_i + \tfrac{1}{2}) < \Gamma(\beta_i + 1) = \beta_i!).$$

The summation over $k$ can be bounded by

$$\sum_{k=0}^{\infty} 2^{-k-1} \frac{(k+d)!}{(k+b)!} = \frac{1}{2} b! \left( 2^d + \binom{d}{b} \right) \leq \frac{1}{2} \left( b! 2^d + 2^b \right) \leq b! 2^d,$$

where the first equality used the identity $\sum_{k=1}^{\infty} 2^{-k} \binom{d+k}{b} = 2^d$. Letting $l := |\boldsymbol{\alpha}|$, we can continue by

$$r_{\boldsymbol{\alpha}}^2 < w_{\boldsymbol{\alpha}}^2 q_{2\boldsymbol{\alpha}} \cdot \frac{b!}{\pi^b d!} b! 2^d = 2^{-l-1} \frac{l!}{\prod_{i=1}^{d} \alpha_i!} \frac{2 \prod_{i=1}^{d} \Gamma\left(\alpha_i + \frac{1}{2}\right)}{V_d \cdot (2l+d) \cdot \Gamma(l+b)} \frac{(b!)^2 2^d}{\pi^b d!}$$

$$\leq 2^{-l+d} \pi^{-2b} \frac{l!(b!)^3}{d!(l+b-1)!(2l+d)} \qquad \left(\text{since } \Gamma\left(\alpha_i + \tfrac{1}{2}\right) < \Gamma(\alpha_i + 1) = \alpha_i!\right)$$

$$\leq 2^{-l+b-1} \pi^{-2b} \binom{l+b}{l}^{-1} \qquad \left(\text{since } \frac{(b!)^2}{d!} \leq 2^{-b}\right).$$

This bound depends on $\boldsymbol{\alpha}$, not directly on $\boldsymbol{\alpha}$. Letting $n_l = \binom{l+d-1}{l}$ and $N_L = \sum_{l=0}^{L} n_l = \binom{d+L}{L}$, it follows that

$$\sum_{l=0}^{L} l n_l = \sum_{l=1}^{L} \frac{l(l+d)!}{d! \cdot l!} = (d+1) \sum_{l=1}^{L} \frac{(l+d)!}{(d+1)!(l-1)!}$$

$$= (d+1) \sum_{l=1}^{L} \binom{l+d}{d+1} = (d+1) \binom{L+d+1}{d+2}.$$

Now we can bound $\lambda_{N_L}$ by

$$\lambda_{N_L}^{N_L} \leq \prod_{i=1}^{N_L} \lambda_i \leq \prod_{l=0}^{L} \left( 2^{-l+b-1} \pi^{-2b} \binom{l+b}{l}^{-1} \right)^{n_l}$$

$$\Rightarrow \quad \log \lambda_{N_L} \leq N_L^{-1} \sum_{l=0}^{L} n_l \left( -(l-b+1) \log 2 - 2b \log \pi - \log \binom{l+b}{l} \right)$$

$$\leq -N_L^{-1} \cdot \log 2 \cdot \sum_{l=0}^{L} l n_l \qquad (\text{since } \log 2 < 2 \log \pi \text{ as the coefficients of } b)$$

$$= -\binom{d+L+1}{d+1}^{-1} \cdot \log 2 \cdot (d+1) \binom{d+L+1}{d+2}$$

$$= -\frac{d+1}{d+2} L \log 2$$

$$\approx -L \log 2$$

$$\Rightarrow \quad \lambda_{N_L} \leq 2^{-L}.$$

This means that the eigenvalue $\lambda_i \leq \varepsilon$ provided that $i \geq N_L$ where $L = \left\lceil \log_2 \frac{1}{\varepsilon} \right\rceil$. Since $N_L \leq d^{L+1}$, that means it suffices to choose $i$ such that

$$i \geq d^{\lceil \log_2 \frac{1}{\varepsilon} \rceil + 1}.$$

This is a quasi-polynomial bound. It seems tight because even in Gaussian RBF kernel, the eigenvalues follow the order of $\lambda_{\boldsymbol{\alpha}} = O(c^{-|\boldsymbol{\alpha}|})$ for some $c > 1$ (Fasshauer & McCourt, 2012, p.A742).

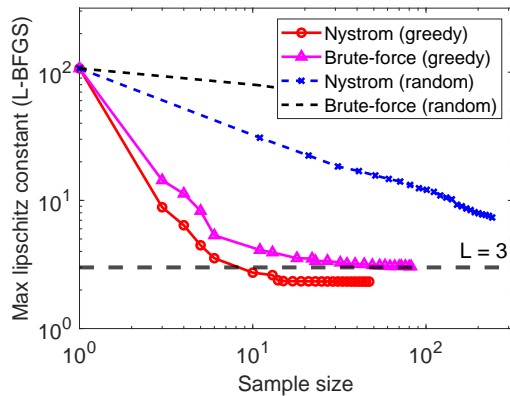

Figure 7: Comparison of efficiency in enforcing Lipschitz constant by various methods

EXPERIMENTS

A.9    EFFICIENCY OF ENFORCING LIPSCHITZ CONSTANT BY DIFFERENT METHODS

The six different ways to train SVMs with Lipschitz regularisation are summarized in Algorithm 1. Figure 7 plots how fast the regularisation on gradient norm becomes effective when more and more points $w$ are added to the constraint set. We call them "samples" although it is not so random in the greedy method, modulo the random initialization of BFGS within the greedy method. The horizontal axis is the loop index $i$ in Algorithm 1, and the vertical axis is $L^{(i)}$ therein, which is the estimation of the Lipschitz constant of the current solution $f^{(i)}$. We used 400 random examples (200 images of digit 1 and 200 images of digit 0) in the MNIST dataset and set $L = 3$ and RKHS norm $\|f\|_{\mathcal{H}} \leq \infty$ for all algorithms. Inverse kernel is used, hence no results are shown for coordinate-wise Nyström.

Clearly the Nyström algorithm is more efficient than the Brute-force algorithm, and the greedy method significantly reduces the number of samples for both algorithms. In fact, Nyström with greedy selection eventually fell below the prespecified $L$, because of the gap in (9).

A.10    EXTENSION TO $\ell_\infty$-NORM ATTACKS FOR OUR KERNEL BASED METHOD

We now extend our kernel based approach to $\ell_\infty$ norm ball attacks. Since most multiclass losses are 1-Lipschitz continuous with respect to $\ell_\infty$ norm on $(f^1(x), \ldots, f^{10}(x))$, we will seek

$$\sup_{x \in X} \sup_{\boldsymbol{u}:\|\boldsymbol{u}\|_\infty \leq 1} \left\| \left[ g^1(x), \ldots, g^{10}(x) \right]^\top \boldsymbol{u} \right\|_\infty \leq L, \quad \text{where} \quad g^c(x) := \nabla f^c(x).$$

The left-hand side (LHS) can be bounded by

$$\text{LHS} = \sup_{x \in X} \max_{1 \leq c \leq 10} \|g^c(x)\|_1 \leq \max_{1 \leq c \leq 10} \sup_{\|\varphi\|_{\mathcal{H}} \leq 1} \left\| G_c^\top \varphi \right\|_1.$$

Given the Nyström approximation $\tilde{G}_c$ of $G_c$, we can enforce the convex constraint of

$$\max_{1 \leq c \leq 10} \sup_{\|\boldsymbol{v}\|_2 \leq 1} \left\| \tilde{G}_c^\top \boldsymbol{v} \right\|_1 \leq L.$$

## A.11 MORE RESULTS ON CROSS-ENTROPY ATTACKS

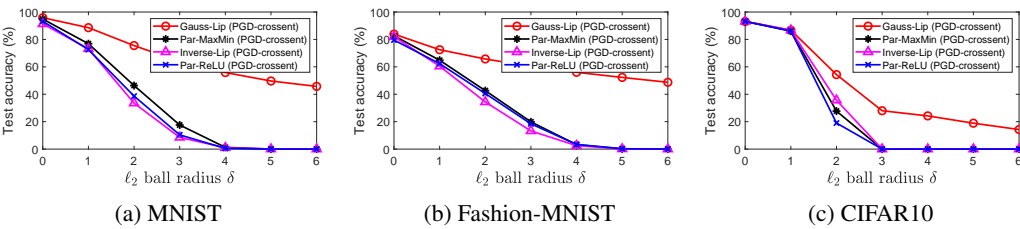

(a) MNIST      (b) Fashion-MNIST      (c) CIFAR10

Figure 8: Test accuracy under PGD attacks on cross-entropy approximation with $\ell_2$ norm bound

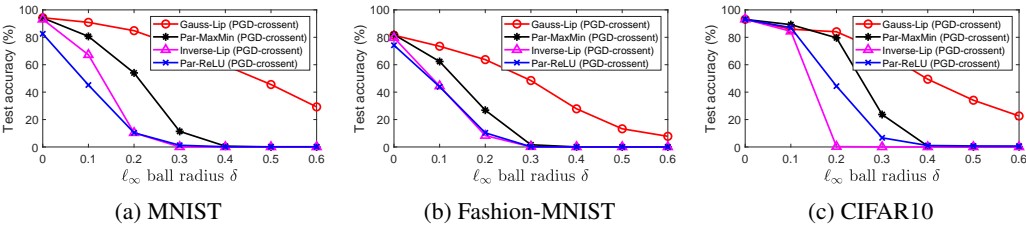

(a) MNIST      (b) Fashion-MNIST      (c) CIFAR10

Figure 9: Test accuracy under PGD attacks on cross-entropy approximation with $\ell_\infty$ norm bound

## A.12 VISUALIZATION OF GRADIENT

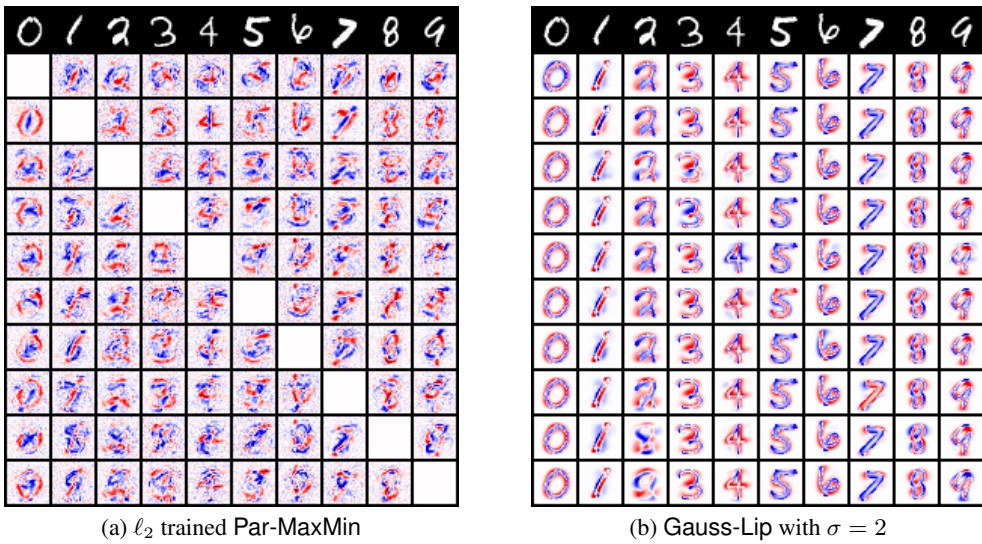

(a) $\ell_2$ trained Par-MaxMin      (b) Gauss-Lip with $\sigma = 2$

Figure 10: Gradients of targeted cross-entropy loss with respect to input images. One image per class (0-9) was sampled randomly from the test set, shown in the first row. A black pixel is encoded by 0, and a white pixel by 1. The 10 rows below show the gradient for different class targets. For example, row 7 column 0 shows the gradient of $f^7$ evaluated at the image of digit 0 shown at the top row. Red and blue stand for positive and negative pixel values, respectively.

A gradient-based attacker tries to decrease the targeted loss by following the negative gradient in Figure 10b, i.e., reduce the pixel value in red area and increase pixel value in blue area.

In order to verify that the robustness of Gauss-Lip is not due to obfuscated gradient, we visualised "large perturbation" adversarial examples, with the $\ell_2$ norm upper bounded by 6. Figure 11 shows how the PGD attacker uses the gradients to perturb the images step by step. At the end of PGD, there are 46 cases where the original image was successfully attacked, i.e., turned into the target class. This is over 50% of the total of 90 cases, and the resulting images look realistic.

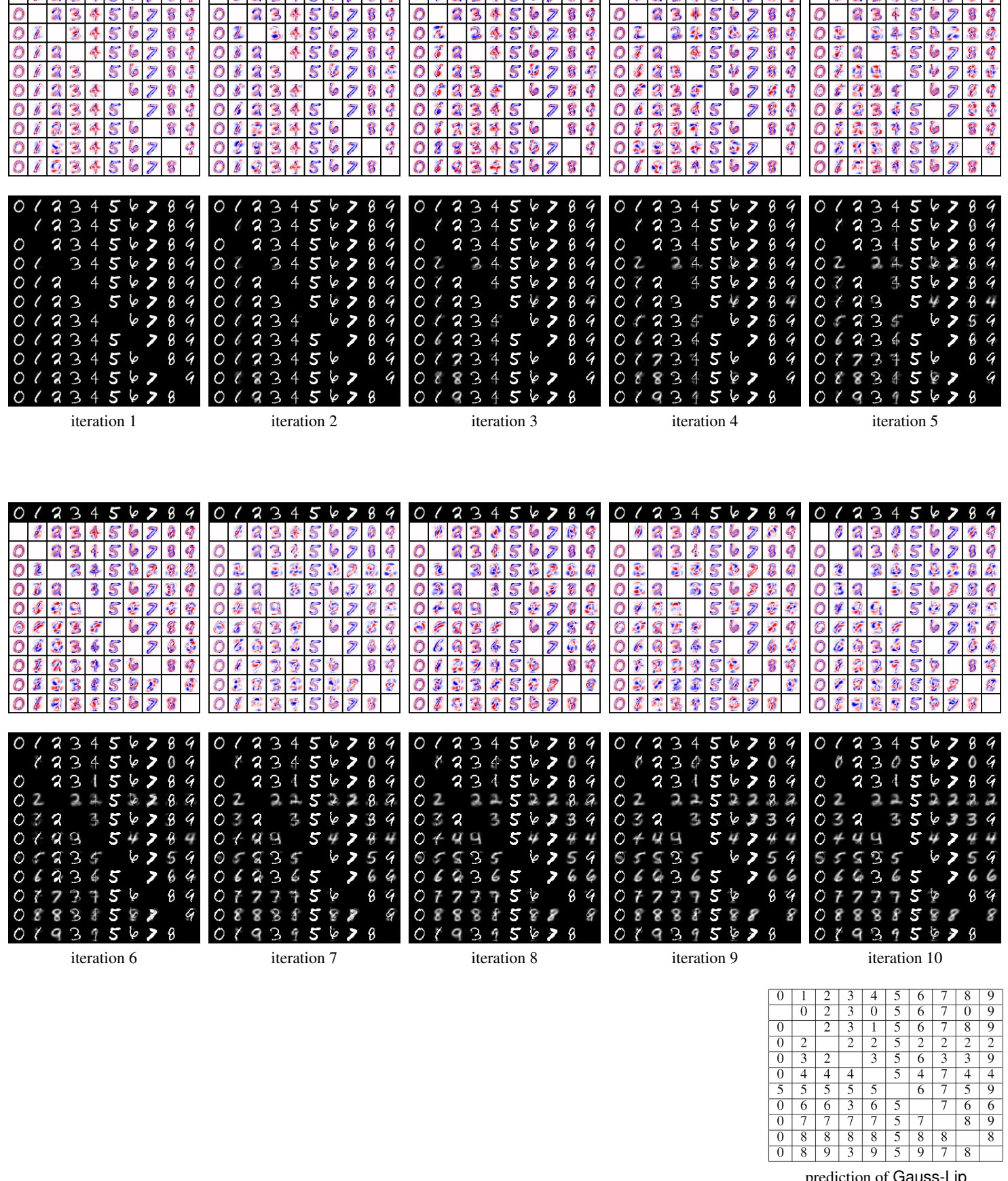

| 0 | 1 | 2 | 3 | 4 | 5 | 6 | 7 | 8 | 9 |
|---|---|---|---|---|---|---|---|---|---|
|   | 0 | 2 | 3 | 0 | 5 | 6 | 7 | 0 | 9 |
| 0 |   | 2 | 3 | 1 | 5 | 6 | 7 | 8 | 9 |
| 0 | 2 |   | 2 | 2 | 5 | 2 | 2 | 2 | 2 |
| 0 | 3 | 2 |   | 3 | 5 | 6 | 3 | 3 | 9 |
| 0 | 4 | 4 | 4 |   | 5 | 4 | 7 | 4 | 4 |
| 5 | 5 | 5 | 5 | 5 |   | 6 | 7 | 5 | 9 |
| 0 | 6 | 6 | 3 | 6 | 5 |   | 7 | 6 | 6 |
| 0 | 7 | 7 | 7 | 7 | 5 | 7 |   | 8 | 9 |
| 0 | 8 | 8 | 8 | 8 | 5 | 8 | 8 |   | 8 |
| 0 | 8 | 9 | 3 | 9 | 5 | 9 | 7 | 8 |   |

prediction of Gauss-Lip

Figure 11: Gradients and perturbed images at each iteration in a 10-step PGD attack using (targeted) cross-entropy approximation, with the $\ell_2$ norm upper bounded by 6. Here the classifier is Gauss-Lip ($\sigma = 2$). The table in the bottom right presents the final predictions of our trained Gauss-Lip on the perturbed images.

| 0 | 1 | 2 | 3 | 4 | 5 | 6 | 7 | 8 | 9 |
|---|---|---|---|---|---|---|---|---|---|
|   | 0 | 0 | 0 | 0 | 0 | 6 | 7 | 0 | 0 |
| 0 |   | 2 | 3 | 1 | 5 | 6 | 7 | 8 | 9 |
| 0 | 2 |   | 2 | 2 | 5 | 2 | 2 | 2 | 2 |
| 3 | 3 | 3 |   | 3 | 3 | 3 | 3 | 3 | 3 |
| 0 | 4 | 4 | 4 |   | 5 | 4 | 4 | 4 | 4 |
| 5 | 5 | 5 | 5 | 5 |   | 5 | 7 | 5 | 5 |
| 6 | 6 | 6 | 6 | 6 | 5 |   | 7 | 6 | 6 |
| 0 | 7 | 7 | 7 | 7 | 5 | 7 |   | 7 | 7 |
| 8 | 8 | 8 | 8 | 8 | 8 | 8 | 8 |   | 8 |
| 0 | 9 | 9 | 9 | 9 | 5 | 9 | 9 | 9 |   |

Figure 12: Left: perturbed images at the end of 100-step PGD attack using (targeted) cross-entropy approximation. Right: classification on the perturbed image given by the trained Gauss-Lip. They are quite consistent with human's perception on the left images.

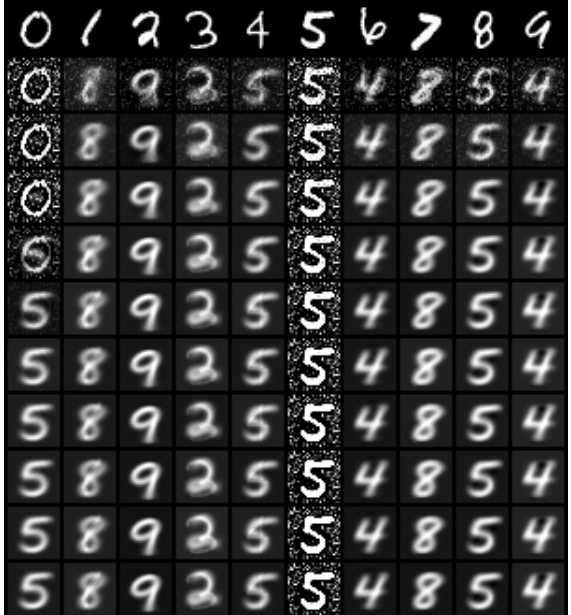

Figure 13: Perturbed images at the end of 100-step PGD attack using (**untargeted**) C&W approximation. The 11 rows show the images after 0, 10, 20, ..., 100 steps of PGD.

To further look into the attack result, we increased PGD to 100 iterations. As shown in Figure 12, now the number of misclassified cases (i.e., unsuccessful attacks that failed to turn an image into the targeted class) drops from 46 to 22, out of 90 cases. The final images are quite realistic. We will further study these remaining cases in the future.

In the above experiments for Figures 11, 12, and 14, PGD was run on the cross-entropy objective. For example, the row corresponding to class 4 tries to promote the likelihood of the target class 4. Naturally the diagonal is not meaningful, hence left empty.

We further ran PGD for 100 iterations on C&W approximation (an untargeted attack used in Figure 3), and the resulting images after every 10 iterations are shown in Figure 13. Here 9 out of 10 images were eventually turned into a different but untargeted class, and the final images are very realistic.

**Another random set of images.** To test if the above result is due to the particularly hard images selected, we randomly selected another set of images and its results for 100-step PGD on cross-entropy objective and C&W objective are shown in Figures 14 and 15, respectively. Interestingly, C&W attack succeeds on all these images, and cross-entropy attack was only unsuccessful in turning 0 into 1.

| 0 | 1 | 2 | 3 | 4 | 5 | 6 | 7 | 8 | 9 |
|---|---|---|---|---|---|---|---|---|---|
|   | 0 | 0 | 0 | 0 | 0 | 0 | 0 | 0 | 0 |
| 0 |   | 1 | 1 | 1 | 1 | 1 | 1 | 1 | 1 |
| 2 | 2 |   | 2 | 2 | 2 | 2 | 2 | 2 | 2 |
| 3 | 3 | 3 |   | 3 | 3 | 3 | 3 | 3 | 3 |
| 4 | 4 | 4 | 4 |   | 4 | 4 | 4 | 4 | 4 |
| 5 | 5 | 5 | 5 | 5 |   | 5 | 5 | 5 | 5 |
| 6 | 6 | 6 | 6 | 6 | 6 |   | 6 | 6 | 6 |
| 7 | 7 | 7 | 7 | 7 | 7 | 7 |   | 7 | 7 |
| 8 | 8 | 8 | 8 | 8 | 8 | 8 | 8 |   | 8 |
| 9 | 9 | 9 | 9 | 9 | 9 | 9 | 9 | 9 |   |

Figure 14: (Another random trial) Left: perturbed images at the end of 100-step PGD attack using (targeted) cross-entropy approximation. Right: classification on the perturbed image given by the trained Gauss-Lip. They are quite consistent with human's perception on the left images.

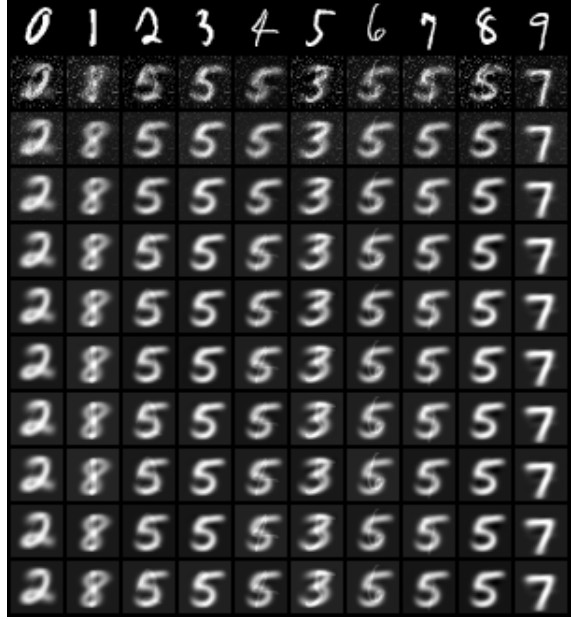

Figure 15: (Another random trial) Perturbed images at the end of 100-step PGD attack using (**untargeted**) C&W approximation. The 11 rows show the images after 0, 10, 20, ..., 100 steps of PGD.

Please note that despite the commonality in using the cross-entropy objective, the setting of targeted attack in Figures 11, 12, and 14 is not comparable to that in Figure 8a, where to enable a batch test mode, an *untargeted* attacker was employed by increasing the cross-entropy loss of the correct class, i.e., decreasing the likelihood of the correct class. This is a common practice.

