# OpenReview forum: "Certifying Distributional Robustness using Lipschitz Regularisation"
_ICLR.cc/2020/Conference — Reject_

### Official Review · AnonReviewer2 · 2019-10-22
**Official Blind Review #2**

**Rating:** 3

**Review:**

In this paper, the authors propose a distributional robust risk minimization method using Lipschitz regularization and give an approach to approximate the Lipschitz constant for product kernels efficiently.

My major concerns are as follows.
1. The result in Theorem 1 is similar to Theorem 3 in [1]. The authors use McShane–Whitney extension theorem [2] to generalize the previous result, which is a trivial derivation.
2. Although the proposed method is theoretically applicable for a wide class of models, it is hard to use in practice, since the Lipschitz constant is too hard to compute exactly. To address the problem, the authors give an approach to approximate the Lipschitz constant for product kernels efficiently. However, the applicability of the method is still limited if it can be used only for product kernels.
3. In the abstract, the authors claim that this paper resolves the previous methods' limitations for deep neural networks. However, there is no corresponding theoretical analysis.
4. Some notations are confusing. For example, in the second paragraph on Page 3, there are two different sets denote $cost_c$; in the definition of $cost_c(\mu,\nu)$, the meaning of the symbol $\Pi$ is unclear; in Theorem 2, the symbol $|\cdot|$ is not defined.

[1] Aman Sinha, Hongseok Namkoong, and John Duchi. Certifying some distributional robustness with principled adversarial  training. In International Conference on Learning Representations (ICLR), 2018.
[2] Iosif Petrakis. Mcshane-Whitney extensions in constructive analysis. arXiv preprint arXiv:1804.06757, 2018.

**Experience Assessment:**

I do not know much about this area.

**Review Assessment: Checking Correctness Of Derivations And Theory:**

I did not assess the derivations or theory.

**Review Assessment: Checking Correctness Of Experiments:**

I did not assess the experiments.

**Review Assessment: Thoroughness In Paper Reading:**

I read the paper at least twice and used my best judgement in assessing the paper.

---

> ### Author Response · Authors · 2019-11-14
> **Rebuttal**
>
> Thank you for your time and review.
>
> 1. Regarding Theorem 1: Since submitting this paper we have been able to derive much stronger results in Theorem 1. These have been added to the revised paper (Section 3), where the key novelties in Theorem 1 are now to
>
> a) leverage a \emph{convexity condition} to bound the tightness of the Lipschitz regularised risk upper bound to the robust risk. In particular, the tightness is characterised in terms of the gap between the loss $\ell_f$ and its convex envelop;
>
> b) prove its tightness for all nonconvex continuous loss functions over the family of Borel measures on a compact set (Proposition 1). This bound on the gap ($\Delta(\mu)$) tightly quantifies the impact of the convexity of parameterised loss on adversarial performance. Such a new guarantee is discovered for the first time in distributionally robust optimisation, where it may serve as a new standard building block for analysing the duality.
>
> Additionally in [1] the authors remark:
>
> "For some simple models, DRO is the same as empirical risk minimization (ERM) with a specific regularizing term. However, for general models, it is less clear how to best perform DRO in practice."
>
> In fact, we are now able to fully characterise the relationship with Theorem 1.  Our results are much more general and stronger than Theorem 3 in [a], and the derivation is not trivial.
>
> 2. Product kernels can induce very rich function spaces. For example, Gaussian kernel is universal [4], meaning that its RKHS is *dense* in the space of continuous functions in the maximum norm.  Lipschitz regularisation in Gaussian RKHS has been limited because it is hard to globally enforce the Lipschitz continuity. Our work is the first that achieves it.
>
> 3. In the abstract, we meant that  "uperbounding with an empirical risk regularised by the Lipschitz constant of the model" is achieved for both "deep neural networks and kernel methods". To avoid this confusion, we modified the sentence to "In this paper we resolve these limitations for a general class of kernel space, and our approach is based on a new upper bound of DRRs using an empirical risk regularised by the Lipschitz constant of the model, e.g., deep neural networks and kernel methods."
>
> 4. Thank you for your observations regarding the notation. We believe these have now been resolved.
>
> [a] Aman Sinha, Hongseok Namkoong, and John Duchi.  "Certifying some distributional robustness with principled adversarial training." In International Conference on Learning Representations (ICLR), 2018.
>
> [1] Matthew Staib and Stefanie Jegelka.  "Distributionally Robust Deep Learning as a Generalization of Adversarial Training."  NIPS Machine Learning and Computer Security Workshop, 2017.
> [2] Jose Blanchet and Karthyek Murthy.  "Quantifying Distributional Model Risk via Optimal Transport." Mathematics of Operations Research 44, no. 2, May 2019.
> [3] Rui Gao and Anton J. Kleywegt.  "Distributionally Robust Stochastic Optimization with Wasserstein Distance." ArXiv:1604.02199 [Math], July 16, 2016.
> [4] Charles A. Micchelli, Yuesheng Xu, and Haizhang Zhang.  "Universal kernels."  Journal of Machine Learning Research (JMLR), 2006.

---

### Official Review · AnonReviewer1 · 2019-10-23
**Official Blind Review #1**

**Rating:** 6

**Review:**

Through the lens of Distributional Robust Risk (DRR), this work draws a link between adversarial robustness and Lipschitz constant regularisation. The authors first provide an upper bound of the DRR (with a Wasserstein ball as the ambiguity set) in terms of the true risk and the Lipschitz constant of the loss function under the current model. They show that the standard adversarial risk can be upper bounded by the DRR, emphasizing that the Lipschitz constant regularised loss can be used as a proxy for adversarially robust training.

The authors then apply this idea to kernel methods and aim to minimise the true risk under a Lipschitz constant constraint. By using a bound between the Lipschitz constant and the largest eigenvalue of the Gram matrix, and by approximating the Gram matrix using Nystrom approximation, they express the Lipschitz constant constraint as a convex constraint. In the case of multiplicative kernels, this approach is shown to be efficient when only using a polynomial number of sample points for the Nystrom approximation.

This method is tested on 3 standard datasets and is shown to outperform state-of-the-art deep learning based approaches over a wide range of perturbation in both l_2 and l_infinity norm.

I have concerns about the novelty of the two main contributions of this work, which are Theorems 1 and 2:
* Theorem 1 is a direct implication of Kantorovich duality, well known in optimal transport.
* Theorem 2 is a rewriting of Proposition 3.1 in [1], which is not cited.

My second concern is that the value of n (number of sample points used in Nystrom approximation) has not been specified for the presented experiments (unless I missed it). This is an important parameter to specify since the algorithm requires to invert an n by n matrix when applying Nystrom approximation, and scalability is usually the main criticism to kernel methods.

When comparing the methods, would it be possible to estimate the Lipschitz constants of the models trained by the various algorithms ? This would enable to observe the true effect of the methods on the Lipschitz constant.

Due to these two concerns above, I tend to reject this submission.

A few comments about clarity:

* It would help to provide the full kernel method algorithm (with more detail than Algorithm 1 in Appendix), e.g., explicit the convex constraint on gamma when training the SVM.

* epsilon notation is used both for the accuracy in the Lipschitz constraint (as in Theorem 3) and the adversarial perturbation (as in equation 3 or Figures 4 and 5). In page 7, paragraph 'Attacks', delta is used once for denoting the perturbation, but epsilon is reused just after. More consistency in the notation would be greatly appreciated.

* It is not clear in the main text that L is a bound on the gradient norms at sample points. It is made clearer in the appendix, so one sentence from the appendix should be added to the main text.


Typos:
* p.6 top: "and analogously for ...(x_1^a,.)" instead of '(x_1^b,.)'

[1] Distributionally Robust Deep Learning as a Generalization of Adversarial Training, M. Staib, S. Jegelka, NIPS, 2017

**Experience Assessment:**

I do not know much about this area.

**Review Assessment: Checking Correctness Of Derivations And Theory:**

I did not assess the derivations or theory.

**Review Assessment: Checking Correctness Of Experiments:**

I assessed the sensibility of the experiments.

**Review Assessment: Thoroughness In Paper Reading:**

I read the paper thoroughly.

---

> ### Author Response · Authors · 2019-11-14
> **Rebuttal**
>
> Thank you for drawing our attention to [1]; we were unaware of this NeurIPS workshop paper and it is a great paper for us to cite.
>
> 1. Regarding Theorem 1:  Since submitting this paper we have been able to derive much stronger results in Theorem 1. These have been added to the revised paper (Section 3), where now the key novelties in Theorem 1 are to
>
> a) leverage a \emph{convexity condition} to bound the tightness of the Lipschitz regularised risk upper bound to the robust risk. In particular, the tightness is characterised in terms of the gap between the loss $\ell_f$ and its convex envelop;
>
> b) prove its tightness for all nonconvex continuous loss functions over the family of Borel measures on a compact set (Proposition 1).  This bound on the gap ($\Delta(\mu)$) tightly quantifies the impact of the convexity of parameterised loss on adversarial performance. Such a new guarantee is discovered for the first time in distributionally robust optimisation, where it may serve as a new standard building block for analysing the duality.
>
> Additionally in [1] the authors remark:
>
> "For some simple models, DRO is the same as empirical risk minimization (ERM) with a specific regularizing term. However, for general models, it is less clear how to best perform DRO in practice."
>
> In fact, we are now able to fully characterise the relationship with Theorem 1.
>
> With regards to the theory of Kantorovich duality, the extent to which it is well-known is only in relation to its utility for formulating the dual of the transportation cost minimisation problem. Its application to distributionally robust risk minimisation is a recent development. We leverage the results of Blanchet & Murthy [2].
>
> 2. Regarding Theorem  2: Proposition 3.1 in [1] as stated is a fairly straight-forward result. However, observe that their equality condition is quite different, and it is not clear at all that it is in fact compatible with the Lipschitz guarantee owing to the use of the (uncommon) $p=\infty$ Wasserstein  metric. Our Theorem 2 also subsumes [3, Cor.~2 (iv)], which is limited to empirical distributions on a finite dimensional space.
>
> 3. The value of $n$ (number of sample points used in Nystr\"om approximation) is less than 200. This posed little computational challenge. We used the greedy scheme to select new sample points; see step 5b of Algorithm 1 in Appendix on page 14.  The whole algorithm converged in 20 iterations (the for loop in steps 2-6), and each iteration added at most 10 new samples.
>
> 4. In the revised paper, we have unfolded the Nystr\"om holistic approach (Step 3.3 of Algorithm 1 in Appendix) below Algorithm 1.  The constraints have been made explicit in the paragraph "Detailed algorithm for multiclass classification".
>
> 5. Thank you for pointing out the use of $\epsilon$ in different contexts.  We have updated the paper. Now $\epsilon$ is used for the accuracy in the Lipschitz constraint, and $\delta$ is used for the budget of adversarial perturbation.  $\delta$ had been used briefly in Theorem 1 to denote probability, but the context is now sufficiently different and self-evident.
>
> 6. Our goal is to ensure $\lambda_{\max} (G^\top G) \le L^2 + \epsilon$ with high probability, and our algorithm is to enforce $\lambda_{\max}(\tilde{P}_G) \le L^2$. This was highlighted at the end of Section 4.1, and is what we did in practice for $\ell_2$ attack in binary classification. The extension to multiclass was detailed in the paragraph "Extending binary kernel machines to multiclass" in Section 5, and extension to $L_\infty$ attack was given in Appendix A.10. Only the sample points $w^s$ were selected greedily through the gradient norm (Step 5b of Algorithm 1), but the gradient norm itself did not serve as a constraint in optimisation.
>
> [1] M. Staib and S. Jegelka.  "Distributionally Robust Deep Learning as a Generalization of Adversarial Training."  NIPS Machine Learning and Computer Security Workshop, 2017.
> [2] Jose Blanchet and Karthyek Murthy.  "Quantifying Distributional Model Risk via Optimal Transport." Mathematics of Operations Research 44, no. 2, May 2019.
> [3] Rui Gao and Anton J. Kleywegt.  "Ditributionally Robust Stochastic Optimization with Wasserstein Distance." ArXiv:1604.02199 [Math], July 16, 2016.

---

### Official Review · AnonReviewer3 · 2019-10-23
**Official Blind Review #3**

**Rating:** 6

**Review:**

Summary:

This paper uses results from distributional robustness to provide bounds of p-norm-constrained adversarial risk which depend on the Lipschitz constant of the underlying classifier. The bulk of the paper focuses on sample-efficient mechanisms to approximate the Lipschitz constants of kernel methods so that a constraint on this Lip constant can be enforced during training. Empirically, the kernel methods are compared to existing deep learning approaches and are shown to be competitive at this scale.

Overview:

This is a technical paper which draws on theoretical results in Kernel methods, distributional robustness, and numerical approximation. I have been unable to verify the correctness of results from all of these areas but to the best of my knowledge the results seem reasonable. Despite the technical depth, the paper is well-written and largely easy to follow.

1) In section 3.1 you should emphasize the threat model more clearly. Distributional robustness may provide tools to evaluate very general forms of adversarial risk but in this work only the p-norm ball threat model is considered. This is a fine restriction and theoretical statements are made correctly but this does not encompass all forms of adversarial risk.

2) I had a hard time understanding Figure 1. The adversarial risk (Eqn 3) is upper-bounded by the variable-radius risk (Eqn 4 LHS) but Figure 1 shows that the adversarial risk of radius $r$ may permit larger perturbations than the variable-radius risk. Could you please explain how this Figure should be interpreted? The caption and main-text description of the Figure are very slim.

3) I like Figures 2 and 3 as they clearly highlight the benefits of both DRR and the proposed approximations in this work. This paper already presents thorough theoretical analysis but a further analysis into the conditions required for $\lambda_{max}(G^T G)$ to guarantee tighter bounds on the Lipschitz constant would be a valuable addition.

4) Unfortunately I was unable to follow the analysis in Section 4.1 due to a lack of prior knowledge. To my understanding, the per-dimensional product factorization allows cheap exact computation of the off-diagonal elements of $G^T G$. This means that only the diagonal elements need be computed by the Nystr\"om approximation. However, I cannot see intuitively how this prevents the sample complexity from scaling with the dimensionality (the number of diagonal elements still grows with $d$).

5) The experimental set-up is mostly standard with the exception that the CIFAR-10 robustness is measured with respect to a learned representation. This is a reasonable setting for validating the theoretical contributions of this work but is not a realistic interpretation of the adversarial threat model considered. The Lipschitz constant of the embedding function (the pre-trained ResNet) is not known and will likely lead to vacuous bounds (for robustness with respect to the input-space).

6) The deep learning experiments utilize the Parseval networks regularization scheme. What is the regularization constant ($\beta$) used for this method? Typically, a small $\beta << 1$ value is chosen which enables easier learning but prevents the Lipschitz constant from being tightly enforced during training. Is the final Lipschitz constant upper-bound computed from the learned weight matrices to ensure that the orthogonality constraint is not violated?

7) There is some disparity between the experimental results presented and the theory in the paper. In particular, this paper explores methods to provide robustness certificates to Lipschitz bounded classifiers but does not compute and validate these certificates empirically.

8) The gradient visualizations were surprising to me. The Par-MaxMin model has interpretable gradients in that the target class is visible through the gradients. However, for the Gauss-Lip model the gradients seem to only depend on the images current class and are suggestive of a form of gradient obfuscation. It may be useful to visualize "large perturbation" adversarial examples for the Gauss-Lip classifier --- what characteristics do these images have? Similarly, one could generate so-called "Distal Adversarials" which are generated by sampling random noise and optimizing them towards a target class. Does the Gauss-Lip model generate realistic looking images or is noise classified accurately?

9) Overall, the experiments seem sufficient to conclude that the Gauss-Lip model is a promising approach to adversarial learning. It is not clear how this method could be scaled to high dimensional data (while there are more obvious avenues for the deep learning alternatives).


Minor:

- In Theorem 1 statement, "If additionally if $\ell_f$ is convex".

**Experience Assessment:**

I have read many papers in this area.

**Review Assessment: Checking Correctness Of Derivations And Theory:**

I assessed the sensibility of the derivations and theory.

**Review Assessment: Checking Correctness Of Experiments:**

I carefully checked the experiments.

**Review Assessment: Thoroughness In Paper Reading:**

I read the paper thoroughly.

---

> ### Author Response · Authors · 2019-11-14
> **Rebuttal**
>
> Thank you for your detailed comments.
>
> 1. Apologies regarding the exposition surrounding Figure 1. The idea is that for the classical adversarial risk, perturbations are only allowed to be up to $r$ distance from each data point. However with the variable-radius risk in Eq 6, the size of the perturbation can vary at each point, so long as the expected perturbation size is bounded by $r$. The explanation in the paper is revised, and to save space, the figure has been moved to Appendix as Figure 5 on page 20.
>
> The motivation for such a risk is not really practical, its use in Theorem 2 is somewhat declarative in that it is "the thing that produces equality" with respect to the distributionally robust risk, and heuristically characterises the upper bound on the Madry et al. adversarial risk. (Although it is also considered by [3, Cor.~2 (iv) ] with some slightly different notation.) In our replies to the other reviewers we mention the extensive progress we were able to make with Theorem 1 since our submission (please feel free to have a look at this). The next theoretical challenge for adversarial learning is addressing the aforementioned gap between the adversarial risk and the variable-radius risk.
>
> 2. It is generally hard to bound the gap between $\sup_{x \in X} \|\nabla f(x)\|^2_2$ and $\lambda_{\max} (G^\top G)$. Empirically we noticed that the gap is smaller when the bandwidth $\sigma$ is larger. This is not surprising because larger $\sigma$ leads to more smooth functions.  To be fair, in Figure 1 and 2, we set $\sigma$ to the median of pairwise distances, which is a common practice in Gaussian kernels.
>
> 3. It is true that only the diagonal entries of $G^\top G$ need to be approximated for product kernels. Although it leads to $d$ entries to be approximated on the diagonal, it is the maximum eigenvalue of $G^\top G$ that is of interest.  Consider a diagonal matrix with entries $\epsilon$ on the diagonal, then its maximum eigenvalue is $\epsilon$. So it is not surprising that the sample complexity in Theorem 3 does not grow linearly in $d$.
>
> 4. The reason why we used a learned embedding for CIFAR-10 is because otherwise neither Parseval network (ReLU or MaxMin) nor our method performs close to the state of the art.  Since the goal of the experiment is to compare the robustness of these two algorithms, basing the comparison on perturbations on a learned representation appears reasonable.
>
> 5. Our experiments exactly followed the code from [1] with $\beta=0.5$, which is equivalent to the first order Bjorck algorithm. We found that the final upper bound of Lipschitz constant computed from the learned weight matrices satisfies the orthogonality constraint as shown in Figure 13 of [2].
>
> 6. We visualised ``large perturbation'' adversarial examples for our Gauss-Lip classifier in Appendix A.12 and found that
>
> a) at the end of 10 steps of PGD, there were 48 cases (combination of target class and starting test image class) where the original image was successfully attacked, i.e., turned into the target class. This was over 50% of the total of 90 cases; and
>
> b) the resulting perturbed images look realistic.
>
> These corroborate that the robustness of Gauss-Lip is not due to obfuscated gradient.  We had been very careful in checking obfuscated gradient, and a discussion was given in the third last paragraph of the paper.
>
> 7. The sample complexity in Theorem 2 is logarithmic in the number of features, and the overall computational cost is therefore almost linear in it.  As a result, Gauss-Lip should be able to scale well to higher dimensional data.
>
> [1] https://github.com/cemanil/LNets
> [2] Cem Anil, James Lucas, and Roger Grosse. “Sorting Out Lipschitz Function Approximation.” International Conference on Machine Learning (ICML), 2019.
> [3] Rui Gao and Anton J. Kleywegt. “Distributionally Robust Stochastic Optimization with Wasserstein Distance.” ArXiv:1604.02199 [Math], July 16, 2016.

---

> > ### Comment · AnonReviewer3 · 2019-11-14
> > **Thank you for your response**
> >
> > Thank you for your detailed response. I am able to follow the clarifications in points 1-3.
> >
> > On the CIFAR-10 learned embeddings: I would argue that this should be clarified in the paper. Especially in the context of your comment 7 above, where you claim that Gauss-Lip should scale well to high-dimensional data. In this case, why is performance on CIFAR-10 data subpar? My original comment 9 referred to the fact that it may be easier to scale the Parseval networks (and other deep-learning) approaches to high dimensional data, even if their current performance on these is lacking. If the authors claim that Gauss-Lip can also be scaled well to these problem sizes then empirical results should be expected.
> >
> > Perhaps I am misunderstanding Figure 11, but I still have concerns with regards to point 6b above. The additional figure does show that on some axes, the gradients do encourage transformations into the target class. However, along many axes (~50% as stated above) the images look relatively unchanged, for example the `5`  and `0` column. In these cases, why do the images look untransformed? Are the gradients in the attack being normalized to ensure that some fixed-size perturbation is added? More specifically, why is the accuracy close to 50% in this case while Fig 3/Fig 4 show near 0 accuracy at a 6-norm ball?

---

> > > ### Author Response · Authors · 2019-11-15
> > > **Thank you for the additional comments**
> > >
> > > 1. We did normalise gradient and fine-tuned the step size. The reason why ~50% images look relatively unchanged is because PGD was not run for long enough. Note in Figure 11 of Appendix A.12, we only ran PGD for 10 steps, while in Figures 3, 4, 8, 9, we ran PGD for 100 steps. To further look into the attack result, we increased PGD to 100 iterations. As shown in Figure 12 of the latest manuscript, now the number of misclassified cases (i.e., unsuccessful attacks that failed to turn an image into the targeted class) drops from 46 to 22, out of 90 cases. The final images are quite realistic. We will further study these remaining cases in the future. Note compared with the last rebuttal, we better tuned step size to improve from 48 to 46.
> > >
> > > In the above experiments for Figures 11, 12, and 14, PGD was run on the cross-entropy objective. For example, the row corresponding to class 4 tries to promote the likelihood of the target class 4. Naturally the diagonal is not meaningful, hence left empty. We further ran PGD for 100 iterations on C&W approximation (an untargeted attack used in Figure 3), and the resulting images after every 10 iterations are shown in Figure 13. Here 9 out of 10 images were eventually turned into a different but untargeted class, and the final images are very realistic.
> > >
> > > ============================
> > > Another random set of images
> > > ============================
> > > To test if the above result is due to the particularly hard images selected, we randomly selected another set of images and its results for 100-step PGD on cross-entropy objective and C&W objective are shown in Figures 14 and 15, respectively. Interestingly, C&W attack succeeds on all these images, and cross-entropy attack was only unsuccessful in turning 0 into 1.
> > >
> > > Please note that despite the commonality in using the cross-entropy objective, the setting of targeted attack in Figures 11, 12, and 14 is not comparable to that in Figure 8a, where to enable a batch test mode, an untargeted attacker was employed by increasing the cross-entropy loss of the correct class, i.e., decreasing the likelihood of the correct class. This is a common practice.
> > >
> > > 2. Scalability is not an issue for Par-MaxMin, Par-ReLU, or Gauss-Lip on CIFAR-10. Since both the code for Par-MaxMin [1] and the paper Anil et al. (2019) [2] only considered fully connected network, we tested CIFAR-10 on it and got about 60% test accuracy.  Anil et al. (2019) [2] did not experiment on CIFAR-10.  This accuracy is consistent with that of Par-ReLU using fully connected nets (Cisse et al., 2017) [4]. We also got a similar accuracy using Gauss-Lip (on clean images). We could compare Gauss-Lip and Par-MaxMin in the same way as for other datasets.  But since the state of the art achieves above 90% accuracy and Par-MaxMin generally outperforms Par-ReLU, it appears fair to focus on the comparison of robustness between Par-MaxMin and Gauss-Lip, both using a residual network as feature embedding.
> > >
> > > [1] https://github.com/cemanil/LNets
> > > [2] Cem Anil, James Lucas, and Roger Grosse. "Sorting Out Lipschitz Function Approximation." International Conference on Machine Learning (ICML), 2019.
> > > [4] Moustapha Cisse, Piotr Bojanowski, Edouard Grave, Yann Dauphin, and Nicolas Usunier. "Parseval networks: Improving robustness to adversarial examples." In International Conference on Machine Learning (ICML), 2017.

---

### Author Response · Authors · 2019-11-15
**General rebuttal**

We thank the reviewers for the comments.  It came to our attention that the "pdfdiff" on OpenReview created a lot of false positive in detecting changes in the revisions.  Indeed, we almost made no change to Section 5 (Experiment), and besides some minor modifications, our major change is to Theorem 1, along with a new lower-bound in Proposition 1. That is mainly in Section 3, before Section 3.1. It allows us to incorporate an important new result on the tightness of our bound. Although it is technically nontrivial, it is not lengthy to interpret. The proof is in the Appendix.

---

### Decision · Program_Chairs · 2019-12-19

**Decision:**

Reject

**Comment:**

This works relates adversarial robustness and Lipschitz constant regularization. After the rebuttal period reviewers still had some concerns. In particular it was felt that Theorem 1 could likely be deduced from known results in optimal transport, and it would be nice to make this connection explicit. There were still concerns about scalability. The authors are encouraged to continue with this work, considering the above points in future revisions.